# GradOrth: A Simple yet Efficient Out-of-Distribution Detection with Orthogonal Projection of Gradients

**Sima Behpour**    **Thang Doan**    **Xin Li**    **Wenbin He**    **Liang Gou**    **Liu Ren**

Bosch Research North America, Bosch Center for Artificial Intelligence (BCAI)
{sima.behpour, thang.doan, xin.li9, wenbin.he2, liang.gou, liu.ren}@us.bosch.com

## Abstract

Detecting out-of-distribution (OOD) data is crucial for ensuring the safe deployment of machine learning models in real-world applications. However, existing OOD detection approaches primarily rely on the feature maps or the full gradient space information to derive OOD scores neglecting the role of **most important parameters** of the pre-trained network over in-distribution (ID) data. In this study, we propose a novel approach called GradOrth to facilitate OOD detection based on one intriguing observation that the important features to identify OOD data lie in the lower-rank subspace of ID data. In particular, we identify OOD data by computing the norm of gradient projection on *the subspaces considered **important** for the in-distribution data*. A large orthogonal projection value (i.e., a small projection value) indicates the sample as OOD as it captures a weak correlation of the ID data. This simple yet effective method exhibits outstanding performance, showcasing a notable reduction in the average false positive rate at a 95% true positive rate (FPR95) of up to 8% when compared to the current state-of-the-art methods.

## 1    Introduction

The issue of identifying out-of-distribution (OOD) data, which falls outside training data distributions, has become a significant focus in deep learning. OOD data challenges real-world model deployment, as it can lead to unreliable or incorrect predictions, particularly in safety-critical applications such as healthcare, autonomous vehicles, and physical sciences [14, 55, 64, 1, 4, 2]. This problem arises because modern Deep Neural Networks (DNNs) produce overconfident predictions on OOD inputs, complicating the separation of in-distribution (ID) and OOD data [63, 41]. The main goal of OOD detection is to develop methods that can accurately detect when a model encounters OOD data, allowing the model to either reject these inputs or provide more informative responses, such as uncertainty indication or confidence measures.

Many studies have investigated approaches to detecting OOD in deep learning [5, 18, 22, 27, 29, 32–34, 38, 39]. The majority of prior work focused on calculating OOD uncertainty from the activation space of a neural network, for example, by using model output [18, 27, 32, 34] or feature representations [29]. Another line of studies like ODIN [32], GradNorm [23], and ExGrad [24] leverage the gradient information of deep neural network models to compute OOD uncertainty score and achieve performant results. GradNorm [23] investigates the richness of the gradient space and presents that gradients provide valuable information for OOD detection. In particular, GradNorm utilizes the vector norm of gradients explicitly as an OOD scoring function. GradNorm, however, considers the full gradient space information, which might be noisy and lead to sub-optimal solutions.

A recent direction of research employs network parameter sparsification to improve OOD detection performance like DICE [47] and ASH [12]. ASH removes a majority of the activation by obtaining the $p$th-percentile of the entire representation. However, the potential consequence may result in

37th Conference on Neural Information Processing Systems (NeurIPS 2023).

diminished performance due to the partial removal of critical parameters within the pre-trained network. Also, this is an empirical approach without a principled way to sparsify models.

Based on key observations presented in *gradient-based* and *sparcification-based* OOD detection methods, an intriguing insight emerges: the crucial discriminative features for OOD data identification reside within the gradient subspace of the ID data. This suggests that by focusing on the gradient information in the subspace of the ID data, which captures the most salient information, we can enhance the accuracy and reliability of OOD data detection algorithms. However, it takes non-trivial work to identify such an efficient gradient subspace.

Inspired by recent low-rank factorization research for DNNs [59, 52, 20, 51], indicating the intrinsic model information resides in a few low-rank dimensions, we introduce a novel approach named **GradOrth**. More specifically, the proposed method, GradOrth, distinguishes OOD samples by employing *orthogonal gradient projection* in the *low-rank subspaces* of ID data (figure 1). These ID subspaces ($S^L$ in figure 1) are derived through singular value decomposition (SVD) of pre-trained network activations, specifically on a small subset of randomly selected ID data. By leveraging SVD, GradOrth effectively computes and identifies the relevant subspaces associated with the ID data, enabling accurate discrimination of OOD samples through orthogonal gradient projection. A large magnitude (figure 1-a) of orthogonal projection (i.e., small projection) serves as a significant criterion for classifying a sample as OOD since it captures a weak correlation with the ID data.

Our key results and contributions are:

- We present GradOrth, a novel and efficient method for out-of-distribution (OOD) detection. Our approach leverages the **most important** parameter space of a pre-trained network and its **gradients** to accomplish this task. To the best of our knowledge, GradOrth is the pioneering endeavor to investigate and showcase the efficacy of the subspace of a DNN's gradients in OOD detection.

- We evaluate the performance of GradOrth on widely-used benchmarks, and it demonstrates competitive results compared to other post-hoc OOD detection baselines. Notably, GradOrth outperforms the strong baseline methods by consistently reducing the false positive rate at the 95th percentile (FPR95) by a margin ranging from 2.71% to 8.05%. Moreover, our experiments highlight that GradOrth effectively enhances OOD detection capabilities while maintaining high accuracy in classifying in-distribution (ID) data.

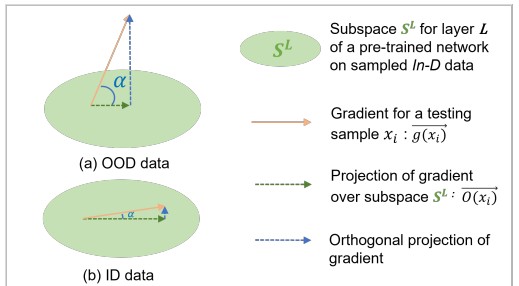

**Figure 1:** The main idea of *GradOrth*: Measuring the orthogonal projection of the gradient of a testing sample onto a $k$-dimension (e.g $k$=2 here) subspace of pre-trained network on ID data, by the angle $\alpha$ between $\overrightarrow{g(x_i)} = \nabla_{\theta^L}\mathcal{L}(\theta^L)$ and $\overrightarrow{O(x_i)} = P_S^L(\nabla_{\theta^L}\mathcal{L}(\theta^L))$. If $\alpha$ is large (i.e., small projection on subspace $\mathcal{S}^L$), shown in (a), the sample $x_i$ is weakly correlated to ID data, and therefore it is recognized as OOD. Otherwise, it is ID data, shown in (b).

- We present a comprehensive analysis, including ablation experiments and theoretical investigation, aimed at enhancing the understanding of our proposed method for OOD detection. Through these rigorous analyses, we aim to provide valuable insights and improve the overall comprehension of the intricacies and effectiveness of our OOD detection approach.

## 2 Background and Our Notations

We consider a neural network with $L$ layers and a set of learning parameters $\theta = \{\theta^l\}_{l=1}^L$ where $\theta^l$ present the learning parameters of layer $l$. $x_i^l$ denotes the **representation of input** $x_i$ at layer $l$ in the successive layers given the data input $x_i$. The network performs the following computation at each layer:

$$x_i^{l+1} = \sigma(f(\theta^l, x_i^l)), \quad l = 1, ...L, \tag{1}$$

where, $\sigma(.)$ is a non-linear function and $f(.,.)$ is a linear function. We leverage matrix notation for input ($\boldsymbol{X}_i$) in convolutional layers and vector notation for input ($x_i$) in fully connected layers. In the first layer, $x_i^1 = x_i$ refers to the raw input data. It is noteworthy to mention that our approach holds applicability across all layers of the network. However, our experimental investigations reveal that the last layer yields the most optimal performance, please refer to appendix, section B for our

empirical studies report. This preference is advantageous as it alleviates significant time complexity arising from gradient computations across multiple network layers.

## 2.1 Input and Gradient Space

Our algorithm capitalizes on the intrinsic property of stochastic gradient descent (SGD) updates lying within the span of input data points, as validated by [62, 20, 60]. The subsequent subsections present this relationship, specifically in fully connected layers. We present the details regarding convolutional layers in the appendix, section A.

**Fully Connected Layer** Consider a single-layer linear neural network in a supervised learning setup where each (input, label) training data pair is driven from a training dataset, $\mathbb{D}$. We use $\boldsymbol{x} \in \mathbb{R}^n$ to present the input vector, $\boldsymbol{y} \in \mathbb{R}^m$ to present the label vector in the dataset, and $\boldsymbol{\theta} \in \mathbb{R}^{m \times n}$ to express the learning parameters (weights) of the network. In general, the network is trained by minimizing a loss function (e.g. mean-squared error) as follows:

$$\mathcal{L} = \frac{1}{2}||\boldsymbol{\theta x} - \boldsymbol{y}||_2^2. \tag{2}$$

Following stochastic gradient optimization, we can present the gradient of this loss with respect to weights as:

$$\nabla_{\boldsymbol{\theta}} \mathcal{L} = (\boldsymbol{\theta x} - \boldsymbol{y})\boldsymbol{x}^T = \boldsymbol{\Omega x}^T, \tag{3}$$

Here, $\boldsymbol{\Omega} \in \mathbb{R}^m$ denotes the error vector. Consequently, the gradient update will reside within the input span ($\boldsymbol{x}$), wherein the elements in $\boldsymbol{\Omega}$ exhibit varying magnitudes, thus influencing the scaling of $\boldsymbol{x}$ accordingly, please refer to section E for the proof. For simplicity, we have considered per-example loss (batch size of 1) and mean-squared loss function here. Furthermore, it is important to mention that the aforementioned relationship remains applicable even in the context of the mini-batch setting or when utilizing alternative loss functions such as cross-entropy loss, where the calculation of $\Omega$ may differ. For more comprehensive information on this subject, please consult the appendix, specifically section D. The input-gradient relationship in equation 3 can be applied to any fully connected layer of a neural network where $\boldsymbol{x}$ is the input to that layer and $\boldsymbol{\Omega}$ is the error coming from the next layer. In addition, this equation also applies to the networks with non-linear units (such as ReLU) and cross-entropy losses, though $\boldsymbol{\Omega}$ will be calculated differently.

## 2.2 Matrix Approximation with SVD:

In our algorithm, we utilize singular value decomposition (SVD) for matrix factorization. Specifically, a rectangular matrix $\boldsymbol{R} = \boldsymbol{U \Sigma V}^T \in \mathbb{R}^{m \times n}$ can be factorized using SVD into the product of three matrices. Here, $\boldsymbol{\Sigma}$ represents a matrix containing the singular values sorted along its main diagonal, while $\boldsymbol{U} \in \mathbb{R}^{m \times m}$ and $\boldsymbol{V} \in \mathbb{R}^{n \times n}$ denote orthogonal matrices [10]. In the case where the rank of the matrix $\boldsymbol{R}$ is $r$ ($r \leq \min(m, n)$), the matrix $\boldsymbol{R}$ can be represented as $\boldsymbol{R} = \sum_{i=1}^{r} \sigma_i \boldsymbol{u}_i \boldsymbol{v}_i^T$, where $\sigma_i \in \text{diag}(\boldsymbol{\Sigma})$ denotes the singular values and $\boldsymbol{u}_i \in \boldsymbol{U}$ as well as $\boldsymbol{v}_i \in \boldsymbol{V}$ represent the left and right singular vectors, respectively. Furthermore, we can formulate the $k$-rank approximation to the matrix as $\boldsymbol{R}_k = \sum_{i=1}^{k} \sigma_i \boldsymbol{u}_i \boldsymbol{v}_i^T$, where $k \leq r$. The specific value of $k$ can be determined as the smallest value that satisfies the condition $||\boldsymbol{R}_k||_F^2 \geq \epsilon_{th}||\boldsymbol{R}||_F^2$. In this equation, $||.||F$ represents the Frobenius norm of the matrix, and $\epsilon_{th}$ ($0 < \epsilon_{th} \leq 1$) serves as the threshold [10]. For a comprehensive explanation of the SVD method, more explanation regarding $k$-rank matrix approximation, and the impact of $\epsilon_{th}$ in method performance please refer to the appendix sections F, H, I, respectively.

## 2.3 Problem Statement: Out-of-distribution Detection

OOD is typically characterized by a distribution that represents unknown scenarios encountered during deployment. These scenarios involve data samples originating from an irrelevant distribution, whose label set has no intersection with the predefined set. Consider the supervised setting where a neural network is given access to a set of training data $\mathbb{D} = \{(\mathbf{x}_i, y_i)\}_{i=1}^N$ drawn from an unknown joint data distribution $P$ defined on $\mathcal{X} \times \mathcal{Y}$ in the training phase. We denote the input space and output space by $\mathcal{X} = \mathbb{R}^n$ and $\mathcal{Y} = \{1, 2, ..., m\}$, respectively.

$$z(\mathbf{x}) = \begin{cases} \text{ID}, & \text{if } O(\mathbf{x}) \geq \gamma \\ \text{OOD}, & \text{if } O(\mathbf{x}) < \gamma. \end{cases} \tag{4}$$

The parameter $\gamma$ is typically selected to ensure a high percentage of correct classification for in-distribution (ID) data, such as 95%. A major hurdle is to establish a scoring function $O(\mathbf{x})$ that

effectively captures the uncertainty associated with OOD samples. Prior approaches have predominantly relied on various factors, including the model's output, gradients, or features, to estimate OOD uncertainty [23, 12]. In our proposed approach, we aim to compute the scoring function $O(\mathbf{x})$ by leveraging *orthogonal gradient projection* on *parameter subspace* of a pre-trained network over ID data. The details of our methodology are described in the subsequent section.

## 2.4 Orthogonal Projection

In this section, we discuss the concept of orthogonal projection and our notation, which holds significant importance in our methodology. To simplify the explanation, we will present it in a 2D space, but it can be extended to higher dimensions. Orthogonal projection serves as a metric that we utilize to calculate the distance between a vector $\overrightarrow{V}$ and a space W (represented as a matrix in this case).

The result of the orthogonal projection of vector $\overrightarrow{V}$ onto space W consists of three essential components: (1) The orthogonal projection vector $\overrightarrow{b}$, (2) the projection vector $\overrightarrow{c}$, and (3) the angle $\alpha$. These three components' values can be utilized to determine the correlation between the vector $\overrightarrow{V}$ and the space W. As depicted in the figure 2, a larger value of $\overrightarrow{b}$ indicates a weaker correlation. On the other hand, $\overrightarrow{c}$ exhibits the opposite pattern, where a larger value of $\overrightarrow{c}$ indicates a stronger correlation. Depending on the specific application requirements, any of these values can be chosen to compute the correlation. In order to align with the OOD score, where a smaller value indicates a higher degree of OODness as presented in equation 4, we incorporate the projection vector ($\overrightarrow{c}$) into our computations.

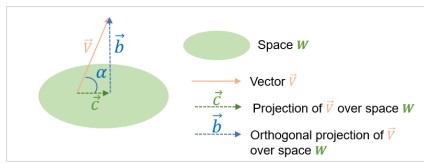

**Figure 2:** Orthogonal Projection

## 3  Our Method: GradOrth

In this section, we describe our method GradOrth where we recognize ID Vs. OOD data from a different view of previous studies, computing the norm of gradient projection on *the subspaces considered **important** for the in-distribution data*.

We define a sample as OOD data if its orthogonal projection value is large (i.e., small projection value), indicating a weak correlation with the ID data.

GradOrth OOD detection is developed following these steps:

1. **Pre-trained Network subspace Computation:** Our $L$-layer neural network with learning parameter $\theta$ is trained using **ID** data. Upon completion of the training process, the model parameters $\theta$ are frozen, resulting in a pre-trained network specialized in ID data. It is worth mentioning that we can also leverage the existing pre-trained network over our interest ID data. To retain the most significant parameters of the pre-trained network with respect to the ID data, we compute the network's last layer ($L$) subspace. For this purpose, we construct a representation matrix denoted as $\boldsymbol{R}_{ID}^L = [\boldsymbol{x}_1^L, \boldsymbol{x}_2^L, ..., \boldsymbol{x}_n^L]$, which concatenates $n$ representations obtained from the network's last layer ($L$) through the forward pass of $n$ randomly selected samples (a small subset, $n \ll N$) of the ID data.

   Next, we perform SVD on $\boldsymbol{R}_{ID}^L$, resulting in $\boldsymbol{R}_{ID}^L = \boldsymbol{U}_{ID}^L \boldsymbol{\Sigma}_{ID}^L (\boldsymbol{V}_{ID}^L)^T$. We then proceed to approximate its rank $k$ by obtaining $(\boldsymbol{R}_{ID}^L)_k$, guided by the given criteria that rely on a specified threshold, denoted as $\epsilon_{th}$:

$$||(\boldsymbol{R}_{ID}^L)_k||_F^2 \geq \epsilon_{th}||\boldsymbol{R}_{ID}^L||_F^2. \tag{5}$$

   The pre-trained network subspace, denoted as $S^L = span\{\boldsymbol{u}_1^L, \boldsymbol{u}_2^L, ..., \boldsymbol{u}_k^L\}$, is defined as the **space of significant representation** for the pre-trained network at the last layer $L$. This subspace is spanned by the first $k$ vectors in $U_{ID}^L$ and encompasses all directions associated with the highest singular values in the representation. We store this subspace, $S^L$, and leverage it in the next step. We present the algorithm to compute the ID subspace in Algorithm 1.

2. **Inference with OOD Data:** During the inference phase, the pre-trained model is exposed to an OOD sample $x_i$. The OOD sample is propagated through the pre-trained network, and subsequently, its gradient at layer L is computed which is presented as $g(x_i) = \nabla_{\boldsymbol{\theta}^L} \mathcal{L}(\boldsymbol{\theta}^L)$.

In accordance with the GradNorm approach, we calculate the cross-entropy loss by comparing the model's predicted softmax probability to a uniform vector used as the target. Consequently, during testing, we employ an all-one vector as the ground truth, assuming a uniform distribution for the target data.

3. **Detector Construction:** The model is transformed into a *detector* by generating a **score** based on its output, enabling the differentiation between ID and OOD inputs. To this end, we compute the norm of sample gradient projection onto the subspace of the pre-trained network $(S)$. We compute projection of the gradients $\nabla_{\boldsymbol{\theta}^L}\mathcal{L}(\boldsymbol{\theta}^L)$ onto the subspace $\mathcal{S}^L$ as follows:

$$P_{S^L}(\nabla_{\boldsymbol{\theta}^L}\mathcal{L}(\boldsymbol{\theta}^L)) = (\nabla_{\boldsymbol{\theta}^L}\mathcal{L}(\boldsymbol{\theta}^L))\mathcal{S}^L(\mathcal{S}^L)'. \tag{6}$$

Here, $(.)'$ presents the matrix transpose. Next, we define the OOD score for the sample as follows by computing the projection norm:

$$O(x_i) = \|P_{S^L}(\nabla_{\boldsymbol{\theta}^L}\mathcal{L}(\boldsymbol{\theta}^L))\| \tag{7}$$

This score serves as a surrogate to characterize the correlation between the sample and ID data that the pre-trained network trained on it. As presented in figure 1, it implies a weak correlation between the new sample $x_i$ and ID when the gradient $g(x_i) = \nabla_{\boldsymbol{\theta}^L}\mathcal{L}(\boldsymbol{\theta}^L)$ has a small projection (large orthogonal projection) onto the subspace of the pre-trained network (large angle $\alpha$) due to the fact that stochastic gradient descent (SGD) updates lie in the span of input data points [65], please refer to the appendix, section E for the proof. Algorithm 2 presents OOD score computation.

---

**Algorithm 1** ID Subspace Computation

1: **function** Compute subspace $(f_{\boldsymbol{\theta}}, \mathcal{D}_{ID}, \epsilon_{th})$
2: //Initialization
3: $S^L \leftarrow [\,]$
4: $B_n \leftarrow$ Sample a mini-batch of size $n$ from ID data $(\mathcal{D}_{ID})$
5: $\boldsymbol{\theta} \leftarrow$ Pre-trained network $f$ learning parameters
6: // subspace Computation
7: $\mathcal{R}_{ID}^L \leftarrow \text{forward}(B_n, f(\boldsymbol{\theta}))$,
8: $\hat{\boldsymbol{U}}_{ID}^L \leftarrow \text{SVD}(\hat{\boldsymbol{R}}_{ID}^L)$
9: $k \leftarrow \text{criteria}(\hat{\boldsymbol{R}}_{ID}^L, \boldsymbol{R}_{ID}^L, \epsilon_{th}^L)$  // Refer to equation 5
10: $\boldsymbol{S}^L \leftarrow \hat{\boldsymbol{U}}_{ID}^L[0:k]$
11: **return** $\boldsymbol{S}^L$

---

**Algorithm 2** OOD Score Computation

1: **function** OOD Detector $(f_{\boldsymbol{\theta}}, \mathcal{D}_{OOD}, \mathcal{S}^L)$
2: //Initialization
3: $\mathcal{O} \leftarrow [\,]$
4: **for** $x_i \in D_{OOD}$ **do**
5: // Gradient computation
6: $\nabla_{\boldsymbol{\theta}}(\mathcal{L}(\boldsymbol{\theta})) \leftarrow \text{SGD}(x_i, f_{\boldsymbol{\theta}})$
7: //Gradient projection computation
8: $P_{S^L}(\nabla_{\boldsymbol{\theta}^L}\mathcal{L}(\boldsymbol{\theta}^L)) \leftarrow (\nabla_{\boldsymbol{\theta}^L}\mathcal{L}(\boldsymbol{\theta}^L))\mathcal{S}^L(\mathcal{S}^L)'$
9: $\mathcal{O}(x_i) \leftarrow \|P_{S^L}(\nabla_{\boldsymbol{\theta}^L}\mathcal{L}(\boldsymbol{\theta}^L))\|$
10: **end for**
11: **return** $\mathcal{O}$

---

# 4 Experiments

In this section, we evaluate the performance of our method *GradOrth* running extensive experiments considering different ID/OOD datasets and network architectures. We follow the experiment setting in general OOD baselines and explain the experimental setup in section 4.1. These empirical studies demonstrate the superior performance of *GradOrth* over existing state-of-the-art baselines that are reported in section 4.1. We report extensive ablations and analyses that provide a deeper understanding of our methodology, please refer to the appendix, section G.

## 4.1 Experimental Setup

**Dataset** We leverage 2 benchmarks proposed by [22] and [12] for detecting OOD images that are based on the large-scale ImageNet dataset and CIFAR dataset. To provide a fair comparison, we adopt an average results-over-5-run approach. In each run, distinct random seeds are employed to select random samples from each class, generating small subsets of in-distribution data. Subsequently, we compute the subspace of the pre-trained network based on these subsets. The OOD scores of the test data are then calculated, and FPR95 and AUROC scores are derived. This process is repeated five times, and the average of these five runs is reported as the final score.

**ImageNet Benchmark:** This benchmark is more challenging than others because it has higher-resolution images and a larger label space of 1,000 categories. To test our approach, we evaluate four OOD test datasets, including subsets of iNaturalist [53], SUN [56], Places [63], and Textures

[8]. These datasets have non-overlapping categories compared to the ImageNet-1k dataset and cover a diverse range of domains including fine-grained, scene, and textural images. We follow the experimental setting reported in [12] and use the Resnet-50 model [16] pre-trained on ImageNet-1k. For a fair comparison, all the methods use the same pre-trained backbone, without regularizing with auxiliary outlier data. Details and hyperparameters of baseline methods can be found in appendix J.1. The outcomes of this study present results obtained by applying GradOrth after the last fully connected layer in all the experiments. In this configuration, the feature size is 2048 for ResNet-50 and 1280 for MobileNetV2. For subspace computation, we choose 10 random samples per class and set the SVD threshold to 0.97.

**CIFAR Benchmark:** We evaluate our approach on the commonly used CIFAR-10 [26], and CIFAR-100 [26] benchmarks as in-distribution data following the experimental setting in [12, 50]. We employ the standard split with 50,000 training images and 10,000 test images. For subspace computation, we choose 5 random samples per class and set the SVD threshold to 0.97. We assess the model on six widely used OOD benchmark datasets: Textures [8], SVHN [40], Places365 [63], LSUN-Crop [61], LSUN-Resize [61], and iSUN [58]. Regarding pre-trained network architecture, we use DenseNet-101 architecture [21]. We leverage pre-trained networks over ID datasets. Please refer to section J.1 in the appendix for more details regarding the experiment setting. It is important to note that no modifications were made to the network parameters during the OOD detection phase.

**Evaluation Metrics** We assess the effectiveness of our proposed method by utilizing threshold-free metrics that are commonly used for evaluating OOD detection, as standardized in [18]. These metrics include (i) AUROC, which stands for the Area Under the Receiver Operating Characteristic curve; and (ii) FPR95, which is the false positive rate. FPR95 represents the probability that a negative (i.e., OOD) example is misclassified as positive (i.e., ID) when the true positive rate is as high as 95 [31].

## 4.2 Results and Discussion

Our experimental studies present the promising performance of OrthoGrad in OOD detection on two benchmarks, ImageNet and CIFAR benchmarks.

### 4.2.1 Experimental Results on Out-of-Distribution Detection

**ImageNet Benchmark:** Our method demonstrates competitive performance, reaching the state-of-the-art level, as indicated in table 1. On the Resnet pre-trained network, GradOrth surpasses ASH-S, ASH-B, and ASH-S by $0.45\%$, $2.47\%$, and $0.93\%$ in terms of FPR95 on the iNaturalist, SUN, and Textures OOD datasets, respectively. When evaluated on the Places OOD dataset, our method achieves an FPR95 of $33.67\%$ and secures the second rank after ASH-B. Furthermore, GradNorm demonstrates an average FPR95 performance of $18.57\%$, outperforming ASH-B by $3.98\%$.

It is important to acknowledge the fact that GradOrth boasts a *low computational complexity*. It only requires computing the subspace of the pre-trained network once and can be conveniently utilized through a simple gradient calculation, without the need for hyper-parameter tuning or additional training during OOD detection. In contrast, certain methods like Mahalanobis [29] require collecting feature representations from intermediate layers for the entire training set, which can be computationally expensive for large-scale datasets like ImageNet. Additionally, GradOrth presents a *stable performance* across most datasets whereas the performance of ASH versions varies across the four OOD datasets. ASH-B outperforms other baselines on the Places dataset but ranks third, second, and third on the other three datasets. A similar pattern is observed for ASH-S in terms of FPR95, where it ranks second, sixth, sixth, and second across the iNaturalist, SUN, Places, and Textures datasets, respectively.

GradOrth also exhibits superior performance in terms of AUROC, outperforming ASH-S by an average of $2.80\%$ across the four datasets. Particularly, GradOrth surpasses ASH-S, ASH-B, and ASH-S by $0.13\%$, $0.66\%$, and $0.46\%$ on the iNaturalist, SUN, and Textures OOD datasets, respectively.

For the pre-trained MobileNet model, our GradOrth approach also demonstrates outstanding performance. We present experimental results on leveraging MobileNet as the pre-trained ID network and evaluate the OOD detection performance on the iNaturalist, SUN, Places, and Textures datasets (the bottom section of table 1). In these experiments, GradOrth demonstrates outstanding performance. In terms of FPR95, GradOrth outperforms ASH-B, DICE+ReAct, DICE+ReAct, and ASH-S by $4.65\%$, $0.40\%$, $6.50\%$, and $0.43\%$, respectively, across the four datasets. Regarding AUROC, GradOrth

outperforms other baselines on average by at least $4.01\%$ and $0.57\%$ in terms of FPR95 and AUROC, respectively.

| Model | Methods | OOD Datasets | | | | | | | | | |
| | | iNaturalist | | SUN | | Places | | Textures | | Average | |
| | | FPR95 ↓ | AUROC ↑ | FPR95 ↓ | AUROC ↑ | FPR95 ↓ | AUROC ↑ | FPR95 ↓ | AUROC ↑ | FPR95 ↓ | AUROC ↑ |
|---|---|---|---|---|---|---|---|---|---|---|---|
| ResNet | Softmax score | 54.99 | 87.74 | 70.83 | 80.86 | 73.99 | 79.76 | 68.00 | 79.61 | 66.95 | 81.99 |
| | ODIN | 47.66 | 89.66 | 60.15 | 84.59 | 67.89 | 81.78 | 50.23 | 85.62 | 56.48 | 85.41 |
| | Mahalanobis | 97.00 | 52.65 | 98.50 | 42.41 | 98.40 | 41.79 | 55.80 | 85.01 | 87.43 | 55.47 |
| | Energy score | 55.72 | 89.95 | 59.26 | 85.89 | 64.92 | 82.86 | 53.72 | 85.99 | 58.41 | 86.17 |
| | GradNorm | 42.46 | 90.33 | 40.73 | 89.96 | 43.48 | 80.64 | 34.48 | 88.43 | 40.29 | 87.34 |
| | ExGrad | 54.11 | 76.91 | 46.73 | 69.74 | 50.62 | 74.27 | 38.12 | 79.37 | 47.40 | 75.07 |
| | ReAct | 20.38 | 96.22 | 24.20 | 94.20 | 33.85 | 91.58 | 47.30 | 89.80 | 31.43 | 92.95 |
| | DICE | 25.63 | 94.49 | 35.15 | 90.83 | 46.49 | 87.48 | 31.72 | 90.30 | 34.75 | 90.77 |
| | DICE + ReAct | 18.64 | 96.24 | 25.45 | 93.94 | 36.86 | 90.67 | 28.07 | 92.74 | 27.25 | 93.40 |
| | VRA-DN | 16.82 | 96.92 | 30.65 | 93.65 | 39.94 | 90.75 | 26.72 | 95.04 | 28.53 | 94.09 |
| | VRA-P | 15.70 | 97.12 | 26.94 | 94.25 | 37.85 | 91.27 | 21.47 | 95.62 | 25.49 | 94.57 |
| | ASH-P | 44.57 | 92.51 | 52.88 | 88.35 | 61.79 | 85.58 | 42.06 | 89.70 | 50.32 | 89.04 |
| | ASH-B | 14.21 | 97.32 | 22.08 | 95.10 | **33.45** | **92.31** | 21.17 | 95.50 | 22.73 | 95.06 |
| | ASH-S | 11.49 | 97.87 | 27.98 | 94.02 | 39.78 | 90.98 | 11.93 | 97.60 | 22.80 | 95.12 |
| | GradOrth (Ours) | **11.04**$^{\pm0.23}$ | **98.00**$^{\pm0.09}$ | **19.61**$^{\pm1.26}$ | **95.76**$^{\pm0.49}$ | 33.67$^{\pm0.18}$ | 91.78$^{\pm0.22}$ | **11.19**$^{\pm0.20}$ | **98.06**$^{\pm0.28}$ | **18.57**$^{\pm0.47}$ | **96.31**$^{\pm0.27}$ |
| MobileNet | Softmax score | 64.29 | 85.32 | 77.02 | 77.10 | 79.23 | 76.27 | 73.51 | 77.30 | 73.51 | 79.00 |
| | ODIN | 55.39 | 87.62 | 54.07 | 85.88 | 57.36 | 84.71 | 49.96 | 85.03 | 54.20 | 85.81 |
| | Mahalanobis | 62.11 | 81.00 | 47.82 | 86.33 | 52.09 | 83.63 | 92.38 | 33.06 | 63.60 | 71.01 |
| | Energy score | 59.50 | 88.91 | 62.65 | 84.50 | 69.37 | 81.19 | 58.05 | 85.03 | 62.39 | 84.91 |
| | ReAct | 42.40 | 91.53 | 47.69 | 88.16 | 51.56 | 86.64 | 38.42 | 91.53 | 45.02 | 89.47 |
| | DICE | 43.09 | 90.83 | 38.69 | 90.46 | 53.11 | 85.81 | 32.80 | 91.30 | 41.92 | 89.60 |
| | DICE + ReAct | 32.30 | 93.57 | 31.22 | 92.86 | 46.78 | 88.02 | 16.28 | 96.25 | 31.64 | 92.68 |
| | ASH-P | 54.92 | 90.46 | 58.61 | 86.72 | 66.59 | 83.47 | 48.48 | 88.72 | 57.15 | 87.34 |
| | ASH-B | 31.46 | **94.28** | 38.45 | 91.61 | 51.80 | 87.56 | 20.92 | 95.07 | 35.66 | 92.13 |
| | ASH-S | 39.10 | 91.94 | 43.62 | 90.02 | 58.84 | 84.73 | 13.12 | 97.10 | 38.67 | 90.95 |
| | GradOrth (Ours) | **26.81**$^{\pm1.19}$ | 93.17$^{\pm0.21}$ | **30.82**$^{\pm0.94}$ | **93.18**$^{\pm0.46}$ | **40.27**$^{\pm1.33}$ | **89.12**$^{\pm0.84}$ | **12.69**$^{\pm0.21}$ | **97.52**$^{\pm0.12}$ | **27.65**$^{\pm0.92}$ | **93.25**$^{\pm0.41}$ |

**Table 1:** OOD detection results with **ImageNet-1k** as ID. GradOrth present outstanding performance on average and across most datasets. We adopted the identical table format and evaluation metrics as introduced in [48, 12]. The ResNet and MobileNet models are pre-trained solely with ID data from the ImageNet-1k dataset. We use ↑ to denote that larger values are preferable, and ↓ to denote that smaller values are preferable. All values are presented as percentages. All values in the table are directly taken from table 1 of [12] except for the gradient-based methods (GradNorm, ExGrad, GradOrth (ours)). For GradNorm and ExGrad, we run this experiment leveraging the code provided by the authors.

**CIFAR Benchmark:** In this research study, we further investigate the performance of GradOrth by conducting additional experimental studies on the CIFAR10 and CIFAR100 datasets. The key observation is that no single method consistently outperforms all other methods across diverse datasets. However, it is noticeable that GradOrth rank is always among the top three across six OOD datasets. This feature presents its promising performance for OOD detection. On the CIFAR10 dataset, GradOrth demonstrates superior performance compared to other baseline methods across six OOD datasets, namely SVHN, LSUN-c, LSUN-r, iSUN, Textures, and Places365. On average, GradOrth outperforms these baselines by $2.71\%$ and $0.32\%$ in terms of FPR95 and AUROC, respectively. Detailed experimental results can be found in table 2. In the LSUN-c OOD dataset, DICE demonstrates superior performance with an impressive $0.26\%$ FPR95, placing it at the top. Our method, on the other hand, ranks second with a respectable $0.81\%$ FPR95. However, the ranking differs when examining the Textures and Places365 datasets. Notably, Gradorth outperforms other baseline methods in both cases, achieving noteworthy FPR95 values of $20.63\%$ and $38.22\%$, respectively. In contrast, DICE attains the sixth and ninth positions in these datasets, displaying comparatively higher FPR95 rates of $41.90\%$ and $48.59\%$.

On the CIFAR100 dataset, GradOrth surpasses its competitors in both FPR95 and AUROC by an average margin of $8.0\%$ and $2.80\%$, respectively, across six well-known OOD datasets. Detailed experimental results are provided in table 3. For a comprehensive discussion and analysis of the CIFAR benchmark, please refer to the appendix, specifically section C.

| Method | OOD Datasets | | | | | | | | | | | | | |
| | SVHN | | LSUN-c | | LSUN-r | | iSUN | | Textures | | Places365 | | Average | |
| | FPR95 ↓ | AUROC ↑ | FPR95 ↓ | AUROC ↑ | FPR95 ↓ | AUROC ↑ | FPR95 ↓ | AUROC ↑ | FPR95 ↓ | AUROC ↑ | FPR95 ↓ | AUROC ↑ | FPR95 ↓ | AUROC ↑ |
|---|---|---|---|---|---|---|---|---|---|---|---|---|---|---|
| Softmax score | 47.24 | 93.48 | 33.57 | 95.54 | 42.10 | 94.51 | 42.31 | 94.52 | 64.15 | 88.15 | 63.02 | 88.57 | 48.73 | 92.46 |
| ODIN | 25.29 | 94.57 | 4.70 | **98.86** | **3.09** | **99.02** | **3.98** | **98.90** | 57.50 | 82.38 | 52.85 | 88.55 | 24.57 | 93.71 |
| Mahalanobis | 6.42 | 98.31 | 56.55 | 86.96 | 9.14 | 97.09 | 9.78 | 97.25 | 21.51 | 92.15 | 85.14 | 63.15 | 31.42 | 89.15 |
| Energy score | 40.61 | 93.99 | 3.81 | 99.15 | 9.28 | 98.12 | 10.07 | 98.07 | 56.12 | 86.43 | 39.40 | 91.64 | 26.55 | 94.57 |
| GradNorm | 18.63 | 94.11 | 1.03 | 99.61 | 3.38 | 98.87 | 36.89 | 91.67 | 50.26 | 89.72 | 50.43 | 84.29 | 26.77 | 93.04 |
| ReAct | 41.64 | 93.87 | 5.96 | 98.84 | 11.46 | 97.87 | 12.72 | 97.72 | 43.58 | 92.47 | 43.31 | 91.03 | 26.45 | 94.67 |
| VRA-P | 18.75 | 96.68 | 1.32 | 99.63 | 5.80 | 98.69 | 5.70 | 98.69 | 34.89 | 93.42 | 39.98 | 91.69 | 17.74 | 96.47 |
| DICE | 25.99 | 95.90 | 0.26 | 99.92 | 3.91 | 98.30 | 4.36 | 97.55 | 41.90 | 93.36 | 48.59 | 89.13 | 20.83 | 95.24 |
| ASH-P | 30.14 | 95.29 | 2.82 | 99.34 | 7.97 | 98.33 | 8.46 | 98.29 | 50.85 | 88.29 | 40.46 | 91.76 | 23.45 | 95.22 |
| ASH-B | 17.92 | 96.86 | 2.52 | 99.48 | 8.13 | 98.54 | 8.59 | 98.45 | 35.73 | 92.88 | 48.47 | 89.93 | 20.23 | 96.02 |
| ASH-S | 6.51 | 98.65 | 0.90 | 99.73 | 4.96 | 98.92 | 5.17 | 98.90 | 24.34 | 95.09 | 48.45 | 88.34 | 15.05 | 96.61 |
| GradOrth | 5.84$^{\pm0.29}$ | 98.72$^{\pm0.08}$ | 0.81$^{\pm0.04}$ | 99.78$^{\pm0.05}$ | 2.33$^{\pm0.07}$ | 98.71$^{\pm0.11}$ | 4.25$^{\pm0.09}$ | 98.32$^{\pm0.57}$ | 20.63$^{\pm1.14}$ | 94.77$^{\pm0.19}$ | 38.22$^{\pm0.38}$ | 91.64$^{\pm0.12}$ | 12.34$^{\pm0.34}$ | 96.99$^{\pm0.19}$ |

**Table 2:** Detailed results on six common OOD benchmark datasets with **CIFAR-10** as ID: Textures [8], SVHN [40], Places365 [63], LSUN-Crop [61], LSUN-Resize [61], and iSUN [58]. GradOrth outperforms other baselines on FPR95 and AUROC on average. For each ID dataset, we use the same DenseNet pre-trained on *CIFAR-10*. We present the first, second, and third ranks in blue, green, and orange colors, respectively. ↑ indicates larger values are better and ↓ indicates smaller values are better.

| Method | OOD Datasets | | | | | | | | | | | | Average | |
|---|---|---|---|---|---|---|---|---|---|---|---|---|---|---|
| | SVHN | | LSUN-c | | LSUN-r | | iSUN | | Textures | | Places365 | | | |
| | FPR95 ↓ | AUROC ↑ | FPR95 ↓ | AUROC ↑ | FPR95 ↓ | AUROC ↑ | FPR95 ↓ | AUROC ↑ | FPR95 ↓ | AUROC ↑ | FPR95 ↓ | AUROC ↑ | FPR95 ↓ | AUROC ↑ |
| Softmax score | 81.70 | 75.40 | 60.49 | 85.60 | 85.24 | 69.18 | 85.99 | 70.17 | 84.79 | 71.48 | 82.55 | 74.31 | 80.13 | 74.36 |
| ODIN | 41.35 | 92.65 | 10.54 | 97.93 | 65.22 | 84.22 | 67.05 | 83.84 | 82.34 | 71.48 | 82.32 | 76.84 | 58.14 | 84.49 |
| Mahalanobis | 22.44 | 95.67 | 68.90 | 86.30 | 23.07 | 94.20 | 31.38 | 93.21 | 62.39 | 79.39 | 92.66 | 61.39 | 55.37 | 82.73 |
| Energy score | 87.46 | 81.85 | 14.72 | 97.43 | 70.65 | 80.14 | 74.54 | 78.95 | 84.15 | 71.03 | 79.20 | 77.72 | 68.45 | 81.19 |
| GradNorm | 31.57 | 93.66 | 9.89 | 96.75 | 58.22 | 87.76 | 59.60 | 84.21 | 59.42 | 88.09 | 57.14 | 82.10 | 45.98 | 88.76 |
| ExGrad | 29.17 | 92.47 | 8.91 | 96.80 | 60.12 | 88.21 | 56.43 | 84.73 | 57.29 | 88.79 | 53.47 | 84.38 | 44.23 | 89.23 |
| ReAct | 83.81 | 81.41 | 25.55 | 94.92 | 60.08 | 87.88 | 65.27 | 86.55 | 77.78 | 78.95 | 82.65 | 74.04 | 62.27 | 84.47 |
| VRA-P | 66.38 | 89.02 | 10.34 | 98.12 | 54.39 | 89.49 | 55.16 | 89.48 | 48.12 | 88.48 | 78.31 | 77.84 | 53.24 | 88.74 |
| DICE | 54.65 | 88.84 | 0.93 | 99.74 | 49.40 | 91.04 | 48.72 | 90.08 | 65.04 | 76.42 | 79.58 | 77.26 | 49.72 | 87.23 |
| ASH-P | 81.86 | 83.86 | 11.60 | 97.89 | 67.56 | 81.67 | 70.90 | 80.81 | 78.24 | 74.09 | 77.03 | 77.94 | 64.53 | 82.71 |
| ASH-B | 53.52 | 90.27 | 4.46 | 99.17 | 48.38 | 91.03 | 47.82 | 91.09 | 53.71 | 84.25 | 84.52 | 72.46 | 48.73 | 88.04 |
| ASH-S | 25.02 | 95.76 | 5.52 | 98.94 | 51.33 | 90.12 | 46.67 | 91.30 | 34.02 | 92.35 | 85.86 | 71.62 | 41.40 | 90.02 |
| GradOrth | 24.27$^{\pm0.33}$ | 93.47$^{\pm1.02}$ | 3.71$^{\pm0.14}$ | 99.07$^{\pm1.04}$ | 48.09$^{\pm0.12}$ | 91.26$^{\pm0.13}$ | 42.73$^{\pm0.59}$ | 91.48$^{\pm0.26}$ | 32.71$^{\pm0.41}$ | 92.62$^{\pm0.11}$ | 48.61$^{\pm0.35}$ | 89.03$^{\pm0.43}$ | 33.35$^{\pm0.32}$ | 92.82$^{\pm0.50}$ |

**Table 3:** Detailed results on six common OOD benchmark datasets with **CIFAR-100** as ID: Textures [8], SVHN [40], Places365 [63], LSUN-Crop [61], LSUN-Resize [61], and iSUN [58]. GradOrth outperforms other baselines on FPR95 and AUROC on average. For each ID dataset, we use the same DenseNet pre-trained on *CIFAR-100*. We present the **first**, **second**, and **third** ranks in blue, green, and orange colors, respectively. ↑ indicates larger values are better and ↓ indicates smaller values are better.

### 4.2.2 Experimental Results on Near-OOD Detection

In this section, we demonstrate the effectiveness of our proposed method on near-OOD detection using competitive deep convolutional neural network architectures such as DenseNet and ResNet on computer vision benchmark datasets: CIFAR-10, CIFAR-100. We follow the experimental setting presented in [29, 46].

| In-dist (model) | OOD | TNR at TPR 95% | AUROC | Detection Acc. |
|---|---|---|---|---|
| | | Baseline [18] / ODIN / Mahalanobis / Gram [45] / GradOrth (Ours) | | |
| CIFAR-10 (ResNet) | CIFAR-100 | 33.3 / 42.0 / 41.6 / 32.9/ **45.3**$^{\pm1.32}$ | 86.4 / 85.8 / 88.2/ 79.0/ **90.3**$^{\pm0.81}$ | 80.4 / 78.6 / 81.2/ 71.7/ **84.6**$^{\pm1.09}$ |
| CIFAR-100 (ResNet) | CIFAR-10 | 19.1 / 18.7 / **20.2** / 12.2 /19.6$^{\pm0.31}$ | 77.1 / 77.2 / **77.5** / 67.9/ 76.9$^{\pm0.93}$ | 71.0 / 71.2 / 72.1/ 63.4/ **75.4**$^{\pm0.53}$ |
| CIFAR-10 (DenseNet) | CIFAR-100 | 40.3 / 53.1 / 14.5 / 26.7 / **58.3**$^{\pm0.26}$ | 89.3 / 90.2 / 58.5 / 72.0 / **94.1**$^{\pm0.31}$ | 82.9 / 82.7 / 57.2 / 67.3 / **84.7**$^{\pm0.62}$ |
| CIFAR-100 (DenseNet) | CIFAR-10 | 18.9 / 16.8 / 7.7 / 10.6 / **21.42**$^{\pm0.52}$ | 75.9 / 74.2 / 60.1 / 64.2 / **77.5**$^{\pm0.37}$ | 69.7 / 68.6 / 57.8 / 60.4 / **72.9**$^{\pm0.68}$ |

**Table 4:** GradOrth presents stable and reliable performance on Near-OOD data setting (CIFAR datasets). This table presents comparison of OOD detection performance for all combinations of model architecture and training datasets following [45].

### 4.2.3 Experimental Results on Semantic Shift versus Non-Semantic Shift

Exploring out-of-distribution data uncovers a compelling challenge: the differentiation between semantic and non-semantic shifts. In pursuit of this investigation, we adopt the experimental methodology outlined in [19] to evaluate the efficacy of the GradOrth approach in this context.

DomainNet [43] presents a collection of high-resolution images, spanning dimensions from 180x180 to 640x880 pixels, distributed across 345 classes within six distinct domains. Notably, four of these domains—real, sketch, infograph, and quickdraw—incorporate class labels, thus facilitating the exploration of various distribution shift scenarios.

Our analysis focused on discerning semantic shifts by segregating classes into two subsets: Split A (comprising classes 0 to 172) and Split B (with classes 173 to 344). In our experiment, real-A served as the in-distribution dataset, while the remaining subsets assumed roles as OOD datasets. Real-B represented a notable example of a semantic shift from real-A, whereas sketch-A exemplified a non-semantic shift. Sketch-B exhibited a convergence of both shift types. The classifier, implemented with a Resnet-34 architecture, underwent a 100-epoch training regimen initiated with a learning rate of 0.01. Images were uniformly center-cropped and resized to dimensions of 224x224 pixels for this experiment.

The results, as depicted in table 5, highlight two prominent trends. Firstly, datasets characterized by both distribution shift types are notably more discernible, with non-semantic shifts following as the next most distinguishable category. Detecting semantic shifts presents the most formidable challenge. Importantly, GradOrth exhibits promising performance, as evidenced by favorable AUROC and TNR@TPR95 metrics, underscoring its robustness and scalability in addressing practical problems. Nevertheless, there remains ample potential for further refinement and enhancement.

**Choice of $L_p$-norm in GradOrth:** In order to investigate the impact of the choice of $L_p$-norm in Equation 7 on OOD detection performance, we conducted an ablation study. Figure 3 illustrates the comparison using $L_{1\sim4}$-norm, $L_\infty$-norm, and the fraction norm (with $p = 0.3$).

Among the different norms considered, we observed that the $L_2$-norm consistently achieved the best OOD detection performance across all four datasets. This finding suggests that the $L_2$-norm, which equally captures information from all dimensions in the gradient space, is better suited for this task. In contrast, higher-order norms tend to unfairly emphasize larger elements over smaller ones due

| OOD | Shift | | AUROC | TNR@TPR95 |
|---|---|---|---|---|
| | S | NS | Baseline [18] / ODIN* / Maha* / DeConf-C [19]*/ GradOrth (ours) | |
| real-B | ✓ | | 75.1 / 69.9 / 53.6 / 69.8 / **79.3** | 15.3 / 15.4 / 5.09 / 14.0/ **17.8** |
| sketch-A | | ✓ | 75.5 / 80.7 / 59.5 / 84.5 / **86.2** | 20.1 / 31.2 / 7.30 / 37.5 / **40.3** |
| sketch-B | ✓ | ✓ | 81.8 / 85.7 / 60.4 / **89.1** / 88.4 | 25.2 / 36.8 / 7.55 / **44.1** / 42.2 |
| infograph-A | | ✓ | 79.6 / 82.7 / 81.5 / 89.0 / **92.1** | 23.5 / 27.8 / 21.6 / 45.4/ **46.8** |
| infograph-B | ✓ | ✓ | 82.1 / 85.3 / 80.9 / 90.9 / **92.6** | 24.8 / 31.7 / 21.9 / / 49.6 / **51.4** |
| quickdraw-A | | ✓ | 78.8 / 96.4 / 67.4 / **96.9** / 96.5 | 21.1 / 79.9 / 3.38 / **83.1** / 80.6 |
| quickdraw-B | ✓ | ✓ | 80.5 / 96.9 / 66.1 / **97.4** / 97.0 | 22.1 / 83.6 / 2.38 / **86.6** / 85.7 |

**Table 5:** GradOrth reliable and stable performance on semantic shift (S) and non-semantic shift settings (NS). Performance of five OOD detection methods using DomainNet. The in-distribution is the real-A subset. Each value is averaged over three runs. All values in the table are directly taken from table 4 of [19] except for GradOrth. The type of distribution shift presents a trend of difficulty to the OOD detection problem: Semantic shift (S) > Non-semantic shift (NS) > Semantic + Non-semantic shift.

to the effect of the exponent $p$. Notably, the $L_\infty$-norm, which only considers the largest element (in absolute value), resulted in the worst OOD detection performance among all the norms tested. Additionally, we evaluated the fraction norm but found that it did not outperform the $L_2$-norm in terms of overall OOD detection performance.

It is worth to mention that while both GradNorm and GradOrth leverage gradients for OOD detection, they adopt differing methodologies. GradNorm places a strong emphasis on the L1 norm, utilizing it to assess gradient magnitudes and discern disparities between ID and OOD data. This focus on gradient magnitude forms the crux of GradNorm's approach. In contrast, GradOrth takes a unique path by employing orthogonal gradient projection within a pre-trained network subspace. In this context, the L2 norm is more suitable for approximating projection length, offering a distinct perspective from the L1 norm employed by GradNorm.

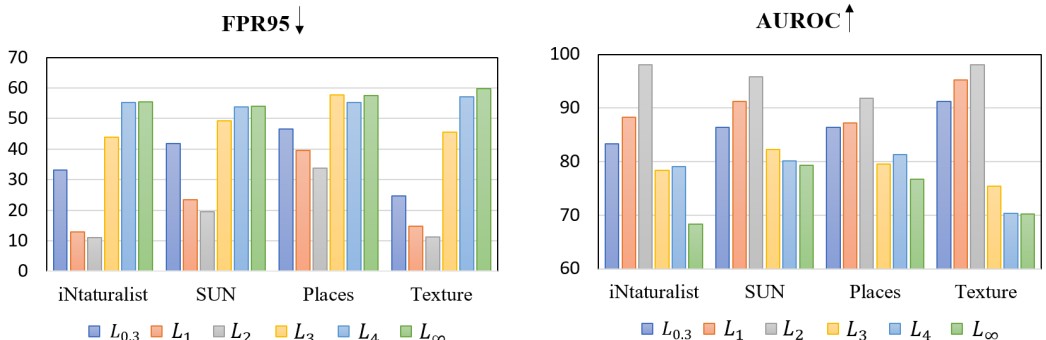

**Figure 3:** Comparison of OOD detection performance using various $L_p$-norms. $L_2$-norm is the best $L$-norm for GradOrth as it provides the lowest FPR95 and largest AUROC for GradOrth. Results are presented for False Positive Rate at 95% True Positive Rate (FPR95) on the left, and Area Under the Receiver Operating Characteristic curve (AUROC) on the right. ↑ indicates larger values are better and ↓ indicates smaller values are better.

## 5 Related Works

To the best of our knowledge, there exists only limited prior research on the use of gradients for detecting OOD inputs. This section aims to explore the connections and differences between our proposed GradOrth method and prior OOD detection approaches that also utilize gradient information. In particular, we will discuss the connection between our approach and the ODIN method, the approach proposed by [28], GradNorm method [23], and ExGrad method [24]. Moreover, [28] utilized gradients from all layers to train a distinct binary classifier, which can lead to a computationally burdensome process for deeper and larger models. Nevertheless, our findings with GradOrth demonstrate that the gradient from the last layer consistently achieves the highest performance compared to other gradient selections. Hence, the computational cost incurred by GradOrth is negligible.

**OOD Gradient-Based Methods** ODIN [32] introduced the concept of utilizing gradient information for OOD detection. Their approach involved a pre-processing technique that added small perturbations obtained from the input gradients. The objective was to enhance the model's confidence in its predictions by increasing the softmax score for each input. This resulted in a larger gap between the softmax scores of ID and OOD inputs, making them more distinguishable and improving OOD detection performance. It is important to note that ODIN indirectly employed gradients through

input perturbation, and the OOD scores were still calculated based on the output space of the perturbed inputs. GradNorm [23] also utilizes gradient information from a neural network to detect distributional shifts between ID and OOD samples. By measuring the norm of gradients with respect to the network's input, it quantifies uncertainty and identifies OOD samples causing significant output changes. ExGrad, proposed by [24], introduces a method akin to GradNorm with two notable distinctions. Firstly, the label distribution of $y$ is derived from the model's predicted distribution ($P$) as opposed to the uniform distribution. Secondly, ExGrad computes the expected norm of the gradient, in contrast to GradNorm which calculates the norm of the expected gradient.

**Discriminative Models for OOD Uncertainty Estimation** The problem of classification with rejection has a long history, dating back to early works on abstention such as [6] and [15], which considered simple model families like SVMs [9]. However, the phenomenon of neural networks' overconfidence in OOD data was not revealed until the work of [41].

Early efforts aimed to improve OOD uncertainty estimation by proposing the ODIN score [32] and Mahalanobis distance-based confidence score [29]. More recently, [34] proposed using an energy score derived from a discriminative classifier for OOD uncertainty estimation, showing advantages over the softmax confidence score both empirically and theoretically. [54] demonstrated that an energy-based approach can improve OOD uncertainty estimation for multi-label classification networks. Additionally, [22] revealed that approaches developed for common CIFAR benchmarks might not effectively translate into a large-scale ImageNet benchmark, highlighting the need to evaluate OOD uncertainty estimation in a large-scale real-world setting. These developments have brought renewed attention to the problem of classification with rejection and the need for effective OOD uncertainty estimation.

**Generative Models for OOD Uncertainty Estimation** Detection of OOD inputs is a crucial problem in machine learning. One popular approach is to use generative models that estimate the density directly. Such models can identify OOD inputs as those lying in low-likelihood regions. To this end, a plethora of literature has emerged to leverage generative models for OOD detection. However, recent studies have shown that deep generative models can assign high likelihoods to OOD data, rendering such models less effective in OOD detection. Additionally, these models can be challenging to train and optimize, and their performance may lag behind their discriminative counterparts. In contrast, our approach relies on a discriminative classifier, which is easier to optimize and achieves stronger performance. While some recent works have attempted to improve OOD detection with generative models using improved metrics, likelihood ratios, and likelihood regret, our approach leverages the energy score from a discriminative classifier and has demonstrated significant advantages over generative models in OOD detection.

**Distributional Shifts** The problem of distributional shift has garnered significant attention in the research community. It is essential to recognize and distinguish between different types of distributional shift problems. In the literature on OOD detection, the focus is typically on ensuring model reliability and detecting label-space shifts [18, 32, 34], where OOD inputs have labels that are disjoint from the ID data, and as such, should not be predicted by the model. On the other hand, some studies have examined covariate shifts in the input space [17, 37, 42], where inputs may be subject to corruption or domain shifts. However, covariate shifts are commonly used to evaluate model robustness and domain generalization performance, where the label space $\mathcal{Y}$ remains the same during test time. It is worth noting that our work focuses on the detection of shifts where the model should not make any predictions, as opposed to covariate shifts where the model is expected to generalize.

## 6 Conclusion

In this paper, we propose GradOrth, a novel OOD uncertainty estimation approach utilizing information extracted from the *important parameter space for ID data* and *gradient space*. Extensive experimental results show that our gradient-based method can improve the performance of OOD detection by up to $8.05\%$ in FPR95 on average, establishing superior performance. We hope that our research brings to light the informativeness of gradient subspace, and inspires future work to utilize it for OOD uncertainty estimation. In our future research, our objective is to investigate GradOrth's capabilities considering different directions like influence functions [25], novelty detection in open-world context [13], data pre-selection [30], and underspecification [36].

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

## Broader Impact

The objective of this research project is to enhance the dependability and safety of contemporary machine learning models. The outcomes of our investigation possess the potential for significant advantages and societal effects, particularly in the realm of safety-critical domains like autonomous driving. It is important to note that our research does not involve any human subjects or breach legal compliance. We do not foresee any possible adverse ramifications resulting from our work. By conducting this study, our intention is to foster increased research efforts and raise awareness within both the research community and society at large regarding the issue of out-of-distribution detection in practical, real-world scenarios.

## Limitation

OOD detection methods may not always detect out-of-distribution samples accurately. As it is presented in our experiments, most OOD detection methods are not able to recognize OOD data across different OOD datasets. They may provide superior performance on some OOD data while catastrophic performance on some other variations of OOD data. Though our method, Gradorth, presents comparable performance to state-of-the-art methods, it still requires more investigation to achieve full stable performance across all OOD datasets.

**Appendix**

The Appendix section is organized as follows:

## A    Input and Gradient Spaces in Convolutional Layers

Our algorithm exploits the observation that updates in stochastic gradient descent (SGD) reside within the subspace spanned by the input data points [62], as discussed in section 2.1 and this appendix, section D. In this section, we aim to establish this relationship specifically for convolutional layers. The analysis presented herein possesses general applicability to any layer within a network, regardless of the task.

In contrast to the weights in a fully connected (FC) layer, filters within a convolutional (Conv) layer operate differently on the input. Let us consider a convolutional layer comprising the input tensor $\mathcal{X} \in \mathbb{R}^{C_i \times h_i \times w_i}$ and filters $\boldsymbol{\theta} \in \mathbb{R}^{C_o \times C_i \times k \times k}$. Their convolution, denoted as $\langle \mathcal{X}, \boldsymbol{\theta}, * \rangle$, yields the output feature map $\mathcal{O} \in \mathbb{R}^{C_o \times h_o \times w_o}$ [35]. Here, $C_i$ ($C_o$) represents the number of input (output) channels in the Conv layer, while $h_i$, $w_i$ ($h_o$, $w_o$) correspond to the height and width of the input (output) feature maps, and $k$ denotes the kernel size of the filters. Figure 4(a) provides a visual representation of this process.

If we reshape $\mathcal{X}$ into a $(h_o \times w_o) \times (C_i \times k \times k)$ matrix denoted as $\boldsymbol{x}$, and reshape $\boldsymbol{\theta}$ into a $(C_i \times k \times k) \times C_o$ matrix denoted as $\boldsymbol{\theta}$, the convolution can be expressed as a matrix multiplication between $\boldsymbol{x}$ and $\boldsymbol{\theta}$, yielding $\mathbf{O} = \boldsymbol{x}\boldsymbol{\theta}$, where $\mathbf{O} \in \mathbb{R}^{(h_0 \times w_0) \times C_o}$.

Formulating the convolution in terms of matrix multiplication provides an intuitive depiction of gradient computation during the back-propagation process. Similar to the fully connected layer scenario, in the convolutional layer, during the backward pass, an error matrix $\boldsymbol{\Omega}$ of size $(h_0 \times w_0) \times C_o$ (equivalent to the size of $\mathbf{O}$) is obtained from the subsequent layer. As illustrated in figure 4(b), the gradient of the loss with respect to the filter weights is computed as follows:

$$\nabla_{\boldsymbol{\theta}} L = \boldsymbol{x}^T \boldsymbol{\Omega}, \tag{8}$$

where $\nabla_{\boldsymbol{\theta}} L$ possesses a shape of $(C_i \times k \times k) \times C_o$ (matching the size of $\boldsymbol{\theta}$). Considering that the columns of $\boldsymbol{x}^T$ correspond to the input.

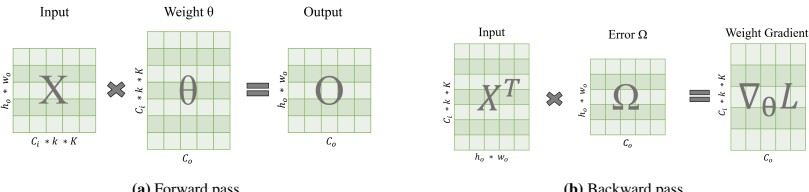

(a) Forward pass                    (b) Backward pass

**Figure 4:** The convolution operation in matrix multiplication format during the forward Pass (a) and backward pass (b).

# B GradOrth Considering All The Network Layers

**This experimental study presents that the gradients obtained from the final layer contain substantial and informative content.** In this experimental study, we aim to investigate the content and significance of gradients obtained from the final layer in a neural network. We explore an alternative variation of the GradOrth method, focusing on the extraction of gradients from all layers of the network. The objective is to analyze the gradients of all trainable parameters across the layers and assess their informativeness. In this paper, we present this version of GradOrth as "GradOrth-All layers" and compare it with the original GradOrth, which utilizes the gradient space from the last layer. To assess the performance of the various gradient spaces, we perform a gradient projection for each layer of the neural network. Subsequently, we calculate the average of these gradient projections across all layers. This procedure allows us to derive an OOD detection score, which we refer to as the OODness score. The experiment is conducted over five random subsets of the ID subspaces. Each subspace is computed by utilizing ten random samples per class in the ImageNet benchmark and five random samples per class in CIFAR benchmarks. We report the average results obtained from these five runs. The outcomes of both GradOrth-All layers and GradOrth-Last layer are presented in tables 6, 7, and 8.

table 6 provides a comparison of the OOD detection performance using the gradient spaces from different layers (all network layers and the last network layer) on two pre-trained network architectures, ResNet and MobileNet, on the ImageNet dataset as ID data. The evaluation metrics used are the False Positive Rate at $95\%$ True Positive Rate (FPR95) and the Area Under the Receiver Operating Characteristic curve (AUROC). These metrics are averaged across four OOD datasets.

Our findings demonstrate that gradients from the last layer consistently outperform gradients from all layers. On ResNet and MobileNet, leveraging the last-layer gradient space results in a $1.57\%$ and $1.52\%$ improvement in FPR95 (on average), respectively, compared to GradOrth utilizing gradients from all network layers.

| Model | Methods | OOD Datasets | | | | | | | | | |
|---|---|---|---|---|---|---|---|---|---|---|---|
| | | iNaturalist | | SUN | | Places | | Textures | | Average | |
| | | FPR95 ↓ | AUROC ↑ | FPR95 ↓ | AUROC ↑ | FPR95 ↓ | AUROC ↑ | FPR95 ↓ | AUROC ↑ | FPR95 ↓ | AUROC ↑ |
| ResNet | GradOrth-All layers | 13.23 | 96.52 | 21.05 | 95.21 | 35.58 | 92.49 | 11.56 | 96.72 | 20.35 | 95.23 |
| | GradOrth-Last layer | 11.04 | 98.00 | 19.61 | 95.76 | 33.67 | 91.78 | 11.19 | 98.06 | 18.57 | 96.31 |
| MobileNet | GradOrth-All layers | 26.14 | 93.83 | 33.28 | 91.38 | 43.71 | 85.37 | 13.61 | 96.87 | 29.17 | 91.86 |
| | GradOrth-Last layer | 26.81 | 93.17 | 30.82 | 93.18 | 40.27 | 89.12 | 12.69 | 97.52 | 27.65 | 93.25 |

**Table 6:** OOD detection results with **ImageNet-1k** as ID. Effect of leveraging all-layers and last-layer gradients space in GradOrth. The OODness score derived from the last layer yields better OOD detection performance mostly. The ResNet and MobileNet models are pre-trained solely with ID data from the ImageNet-1k dataset. We use ↑ to denote that larger values are preferable, and ↓ to denote that smaller values are preferable. All values are presented as percentages.

In the pursuit of an extensive investigation, the CIFAR benchmark is taken into account to examine the impact of various network gradient spaces, namely the last network layer and all network layers, on the performance of GradOrth. The findings, depicted in tables 7 and 8, reveal that utilizing last layer gradients in GradOrth yields superior results compared to employing gradients from all network layers, with improvements of $0.82\%$ and $1.49\%$ in FPR95 on average, respectively.

This observed outcome is highly advantageous, as gradients calculated with respect to deeper layers demonstrate computational efficiency when compared to utilizing gradients from all layers. Remarkably, the GradOrth variant derived from the final linear layer exhibits the most favorable outcomes. From a practical perspective, it is only necessary to perform back-propagation with respect

to the last linear layer, resulting in minimal computational overhead. Therefore, our primary findings are predicated on the utilization of the last fully connected (FC) layer within the neural network.

| Method | OOD Datasets | | | | | | | | | | | | Average | |
| | SVHN | | LSUN-c | | LSUN-r | | iSUN | | Textures | | Places365 | | | |
| | FPR95 ↓ | AUROC ↑ | FPR95 ↓ | AUROC ↑ | FPR95 ↓ | AUROC ↑ | FPR95 ↓ | AUROC ↑ | FPR95 ↓ | AUROC ↑ | FPR95 ↓ | AUROC ↑ | FPR95 ↓ | AUROC ↑ |
|---|---|---|---|---|---|---|---|---|---|---|---|---|---|---|
| GradOrth-All layers | 7.32 | 98.24 | 1.04 | 99.68 | 4.11 | 98.47 | 4.10 | 98.36 | 24.52 | 93.06 | 37.91 | 92.11 | 13.16 | 96.65 |
| GradOrth-Last layer | 5.84 | 98.72 | 0.81 | 99.78 | 2.33 | 98.71 | 4.25 | 98.32 | 20.63 | 94.77 | 38.22 | 91.64 | 12.34 | 96.99 |

**Table 7:** GradOrth leveraging last-layer network's gradient space outperforms Gradorth leveraging all network layers gradient space on CIFAR10 dataset. Detailed results on six common OOD benchmark datasets with **CIFAR-10** as ID: Textures [8], SVHN [40], Places365 [63], LSUN-Crop [61], LSUN-Resize [61], and iSUN [58]. . For each ID dataset, we use the same DenseNet pre-trained on *CIFAR-10*. ↑ indicates larger values are better and ↓ indicates smaller values are better.

| Method | OOD Datasets | | | | | | | | | | | | Average | |
| | SVHN | | LSUN-c | | LSUN-r | | iSUN | | Textures | | Places365 | | | |
| | FPR95 ↓ | AUROC ↑ | FPR95 ↓ | AUROC ↑ | FPR95 ↓ | AUROC ↑ | FPR95 ↓ | AUROC ↑ | FPR95 ↓ | AUROC ↑ | FPR95 ↓ | AUROC ↑ | FPR95 ↓ | AUROC ↑ |
|---|---|---|---|---|---|---|---|---|---|---|---|---|---|---|
| GradOrth-All layers | 27.61 | 92.84 | 4.32 | 97.63 | 49.27 | 89.48 | 44.07 | 90.17 | 34.86 | 91.25 | 48.92 | 88.83 | 34.84 | 91.70 |
| GradOrth-Last layer | 24.27 | 93.37 | 3.71 | 99.07 | 48.09 | 91.26 | 42.73 | 91.48 | 32.71 | 92.62 | 48.61 | 89.03 | 33.35 | 92.82 |

**Table 8:** GradOrth leveraging the last-layer network's gradient space outperforms Gradorth leveraging all network layers' gradient space on the CIFAR100 dataset. Detailed results on six common OOD benchmark datasets with **CIFAR-100** as ID: Textures [8], SVHN [40], Places365 [63], LSUN-Crop [61], LSUN-Resize [61], and iSUN [58]. For each ID dataset, we use the same DenseNet pre-trained on *CIFAR-100*. ↑ indicates larger values are better and ↓ indicates smaller values are better.

## C   Analysis of the Number of ID Samples

The initial step in our proposed method involves computing the subspace of the pre-trained neural network using the ID data. To accomplish this, we utilize a small number of data samples and pass them through the forward pass of the pre-trained network, without altering the learned parameters. Subsequently, we compute the subspace based on the last layer of the network.

To ensure a comprehensive study, we conduct an empirical investigation and compute variations of subspaces by considering different numbers of samples per class. Specifically, we vary the number of samples from 5 to 40 and 10 to 30 in the CIFAR and ImageNet1K benchmarks, respectively. For each variation, we compute five random subspaces and report the average results obtained from these five subspaces. The experimental results obtained from our study are presented in tables 9 and 10. Specifically, we introduce a notation to describe the experiments using GradOrth on the pre-trained network subspace computed based on a specific number of samples per class. We denote this notation as GradOrth-$S_n$, where $n$ represents the number of samples per class used to compute the subspace.

| Model | OOD Datasets | | | | | | | | | | | | Average | |
| | SVHN | | LSUN-c | | LSUN-r | | iSUN | | Textures | | Places365 | | | |
| | FPR95 ↓ | AUROC ↑ | FPR95 ↓ | AUROC ↑ | FPR95 ↓ | AUROC ↑ | FPR95 ↓ | AUROC ↑ | FPR95 ↓ | AUROC ↑ | FPR95 ↓ | AUROC ↑ | FPR95 ↓ | AUROC ↑ |
|---|---|---|---|---|---|---|---|---|---|---|---|---|---|---|
| GradOrth-$S_5$ | 5.84 | 98.72 | 0.81 | 99.78 | 2.33 | 98.71 | 4.25 | 98.32 | 20.63 | 94.77 | 38.22 | 91.64 | 12.34 | 96.99 |
| GradOrth-$S_{10}$ | 5.80 | 98.74 | 0.75 | 99.79 | 2.34 | 98.69 | 4.11 | 98.41 | 20.37 | 94.82 | 38.13 | 92.01 | 11.86 | 97.07 |
| GradOrth-$S_{20}$ | 5.61 | 98.80 | 0.69 | 99.81 | 2.27 | 98.74 | 4.17 | 98.35 | 20.59 | 94.79 | 38.09 | 91.71 | 11.90 | 97.03 |
| GradOrth-$S_{40}$ | 5.62 | 98.79 | 0.69 | 99.83 | 2.24 | 98.75 | 4.16 | 98.38 | 20.55 | 94.84 | 38.11 | 91.74 | 11.90 | 97.05 |

**Table 9:** Detailed results on six common OOD benchmark datasets considering different numbers of ID samples (per class) in subspace computation. For each ID dataset, we use the same DenseNet pre-trained on *CIFAR-10*. ↑ indicates larger values are better and ↓ indicates smaller values are better.

| Model | OOD Datasets | | | | | | | | | | | | Average | |
| | SVHN | | LSUN-c | | LSUN-r | | iSUN | | Textures | | Places365 | | | |
| | FPR95 ↓ | AUROC ↑ | FPR95 ↓ | AUROC ↑ | FPR95 ↓ | AUROC ↑ | FPR95 ↓ | AUROC ↑ | FPR95 ↓ | AUROC ↑ | FPR95 ↓ | AUROC ↑ | FPR95 ↓ | AUROC ↑ |
|---|---|---|---|---|---|---|---|---|---|---|---|---|---|---|
| GradOrth-$S_5$ | 24.27 | 93.37 | 3.71 | 99.07 | 48.09 | 91.26 | 42.73 | 91.48 | 32.71 | 92.62 | 48.61 | 89.03 | 33.35 | 92.82 |
| GradOrth-$S_{10}$ | 24.25 | 93.38 | 3.73 | 99.09 | 48.08 | 91.29 | 42.70 | 91.52 | 32.65 | 92.66 | 48.63 | 89.04 | 33.34 | 92.83 |
| GradOrth-$S_{20}$ | 24.21 | 93.40 | 3.70 | 99.10 | 48.04 | 91.33 | 42.69 | 91.50 | 32.64 | 92.65 | 48.54 | 89.05 | 33.30 | 92.83 |
| GradOrth-$S_{40}$ | 24.09 | 93.46 | 3.67 | 99.11 | 47.86 | 91.38 | 42.53 | 91.73 | 32.49 | 92.72 | 46.91 | 90.12 | 32.92 | 93.09 |

**Table 10:** Detailed results on six common OOD benchmark datasets considering different numbers of ID samples (per class) in subspace computation. For each ID dataset, we use the same DenseNet pre-trained on *CIFAR-100*. ↑ indicates larger values are better and ↓ indicates smaller values are better.

The experimental results displayed in tables 9 and 10 indicate that increasing the number of samples per class during the computation of the subspace for the pre-trained network does not have a significant impact on the overall performance of OOD detection. This observation suggests that leveraging a pre-trained network, which has already learned the data well, diminishes the influence of the number of samples per class in the subspace computation.

| Model | Methods | OOD Datasets | | | | | | | | | |
|---|---|---|---|---|---|---|---|---|---|---|---|
| | | iNaturalist | | SUN | | Places | | Textures | | Average | |
| | | FPR95 ↓ | AUROC ↑ | FPR95 ↓ | AUROC ↑ | FPR95 ↓ | AUROC ↑ | FPR95 ↓ | AUROC ↑ | FPR95 ↓ | AUROC ↑ |
| ResNet | GradOrth-$S_{10}$ | $11.04^{\pm0.23}$ | $98.00^{\pm0.09}$ | $19.61^{\pm1.26}$ | $95.76^{\pm0.49}$ | $33.67^{\pm0.18}$ | $91.78^{\pm0.22}$ | $11.19^{\pm0.20}$ | $98.06^{\pm0.28}$ | $18.88^{\pm0.47}$ | $95.90^{\pm0.27}$ |
| | GradOrth-$S_{20}$ | $10.98^{\pm0.21}$ | $98.08^{\pm0.14}$ | $19.63^{\pm1.11}$ | $95.78^{\pm0.32}$ | $33.63^{\pm0.26}$ | $91.80^{\pm0.18}$ | $11.20^{\pm0.22}$ | $98.05^{\pm0.25}$ | $18.86^{\pm0.45}$ | $95.92^{\pm0.22}$ |
| | GradOrth-$S_{30}$ | $11.00^{\pm0.27}$ | $98.04^{\pm0.12}$ | $19.66^{\pm0.51}$ | $95.78^{\pm0.27}$ | $33.68^{\pm0.26}$ | $91.83^{\pm0.23}$ | $11.17^{\pm0.25}$ | $98.08^{\pm0.19}$ | $18.88^{\pm0.31}$ | $95.93^{\pm0.26}$ |
| MobileNet | GradOrth-$s_{10}$ | $26.81^{\pm1.19}$ | $93.17^{\pm0.21}$ | $30.82^{\pm0.94}$ | $93.18^{\pm0.46}$ | $40.27^{\pm1.33}$ | $89.12^{\pm0.84}$ | $12.69^{\pm0.21}$ | $97.52^{\pm0.12}$ | $27.65^{\pm0.92}$ | $93.25^{\pm0.41}$ |
| | GradOrth-$s_{20}$ | $26.78^{\pm0.83}$ | $93.20^{\pm0.24}$ | $30.76^{\pm0.70}$ | $93.20^{\pm0.39}$ | $40.22^{\pm1.16}$ | $89.13^{\pm0.69}$ | $12.65^{\pm0.24}$ | $97.55^{\pm0.17}$ | $27.60^{\pm0.73}$ | $93.27^{\pm0.37}$ |
| | GradOrth-$s_{30}$ | $26.75^{\pm0.91}$ | $93.23^{\pm0.20}$ | $30.75^{\pm0.82}$ | $93.22^{\pm0.45}$ | $40.28^{\pm1.21}$ | $89.10^{\pm0.72}$ | $12.66^{\pm0.19}$ | $97.54^{\pm0.20}$ | $27.61^{\pm0.78}$ | $93.27^{\pm0.39}$ |

**Table 11:** GradOrth performance is not impacted by the number of ID samples significantly. Detailed results on four common OOD benchmark datasets considering different numbers of ID samples (per class) in subspace computation. We use ResNet and MobileNet pre-trained on *ImageNet-1k*. In our $s_n$ notation, $n$ represents the number of ID samples per class.

# D   Cross-Entropy Loss and its Derivative

In this section, we provide a detailed explanation of the cross-entropy loss and its derivative.

Consider a single-layer linear neural network in a supervised learning setting, where each training data pair $(\boldsymbol{x}, \boldsymbol{y})$ is drawn from a training dataset $\mathbb{D}$. Here, $\boldsymbol{x} \in \mathbb{R}^n$ represents the input vector, $\boldsymbol{y} \in \mathbb{R}^m$ represents the label vector in the dataset, and $\boldsymbol{\theta} \in \mathbb{R}^{m \times n}$ represents the learning parameters (weights) of the network. The model's prediction on input $\boldsymbol{x}$ is denoted by $f(\boldsymbol{x}; \boldsymbol{\theta})$. For classification problems, $f(\boldsymbol{x}; \boldsymbol{\theta}) = \boldsymbol{\theta}\boldsymbol{x}$, where $f_k(\boldsymbol{x}; \boldsymbol{\theta})$ represents the $k$-th logit associated with the $k$-th class.

The total loss on the training set (empirical risk) is denoted by:

$$L_{\mathbb{D}}(\boldsymbol{\theta}) = \sum_{(\boldsymbol{x}, \boldsymbol{y}) \in \mathbb{D}} L_{(\boldsymbol{x}, \boldsymbol{y})}(\boldsymbol{\theta}), \tag{9}$$

where the per-example loss is defined as:

$$L_{(\boldsymbol{x}, \boldsymbol{y})}(\boldsymbol{\theta}) = \ell(\boldsymbol{y}, f(\boldsymbol{x}; \boldsymbol{\theta})), \tag{10}$$

and $\ell(\cdot, \cdot)$ represents a differentiable non-negative loss function.

For classification problems, the softmax cross-entropy loss is commonly used, given by:

$$\ell(\boldsymbol{y}, f(\boldsymbol{x}; \boldsymbol{\theta})) = -\sum_{k=1}^{m} y_k \log a_k, \tag{11}$$

where $a_k = \exp(f_k(\boldsymbol{x}; \boldsymbol{\theta})) / \sum_k \exp(f_k(\boldsymbol{x}; \boldsymbol{\theta}))$ represents the softmax output for the $k$-th class.

Using the chain rule, the gradient of the loss can be expressed as:

$$\nabla L_{(\boldsymbol{x}, \boldsymbol{y})}(\boldsymbol{\theta}) = \nabla f(\boldsymbol{x}; \boldsymbol{\theta}) \ell'(\boldsymbol{y}, f(\boldsymbol{x}; \boldsymbol{\theta})), \tag{12}$$

where $\ell'(\cdot, \cdot) \in \mathbb{R}^m$ denotes the derivative of $\ell(\cdot, \cdot)$ with respect to its second argument, and $\nabla f(\boldsymbol{x}; \boldsymbol{\theta})$ represents the gradient of the model $f$ with respect to its second argument (i.e., the parameters). For the classification problem with cross-entropy softmax loss, we have:

$$\nabla f(\boldsymbol{x}; \boldsymbol{\theta}) = [\nabla f_1(\boldsymbol{x}; \boldsymbol{\theta}); \ldots; \nabla f_m(\boldsymbol{x}; \boldsymbol{\theta})], \tag{13}$$

where $\nabla f_k(\boldsymbol{x}; \boldsymbol{\theta}) \in \mathbb{R}^m$ represents the gradient of the $k$-th logit with respect to the parameters. For simplicity, let's write equation 13 as:

$$\nabla f(\boldsymbol{x}; \boldsymbol{\theta}) = \boldsymbol{x}^T. \tag{14}$$

Considering that the derivative of the loss is given by:

$$\ell'(\boldsymbol{y}, f(\boldsymbol{x}; \boldsymbol{\theta})) = [a_1 - y_1, \ldots, a_m - y_m]^\top = \rho, \tag{15}$$

Here, $\rho$ denotes the error vector. Considering equations 12, 14, and 15, we can write equation 12 as:

$$\nabla L_{(\boldsymbol{x},\boldsymbol{y})}(\boldsymbol{\theta}) = \rho \boldsymbol{x}^T, \tag{16}$$

As a result, the gradient update will be confined within the input span ($\boldsymbol{x}$), where the elements in $\boldsymbol{\rho}$ display heterogeneous magnitudes, thereby impacting the scaling of $\boldsymbol{x}$ correspondingly.

## E    Gradient Span Proof

Considering that the batch loss is the summation of losses incurred by individual examples, the overall batch loss for $n$ samples can be represented as:

$$L_{\text{batch}} = \sum_{i=1}^{n} L_i, \tag{17}$$

where $L_i$ represents the loss of sample $(x_i, y_i)$.

When employing the mean-squared error loss function, the loss of a batch is calculated as the sum of the losses of individual samples, which can be expressed as:

$$L_{\text{batch}} = \sum_{i=1}^{n} L_i = \sum_{i=1}^{n} \frac{1}{2} ||\boldsymbol{\theta} \boldsymbol{x}_i - \boldsymbol{y}_i||_2^2. \tag{18}$$

Following stochastic gradient optimization, we can present the gradient of this loss (per sample) with respect to weights as:

$$\nabla_{\boldsymbol{\theta}} \mathcal{L} = (\boldsymbol{\theta} \boldsymbol{x} - \boldsymbol{y}) \boldsymbol{x}^T = \boldsymbol{\Omega} \boldsymbol{x}^T, \tag{19}$$

Here, $\boldsymbol{\Omega} \in \mathbb{R}^m$ denotes the error vector. Therefore, the gradient of the batch loss with respect to the weights can be expressed as:

$$\nabla_{\boldsymbol{\theta}} L_{\text{batch}} = \boldsymbol{\Omega}_1 \boldsymbol{x}_1^T + \boldsymbol{\Omega}_2 \boldsymbol{x}_2^T + \ldots + \boldsymbol{\Omega}_n \boldsymbol{x}_n^T. \tag{20}$$

It is noteworthy that the gradient update remains confined within the subspace spanned by the $n$ input examples.

Considering cross-entropy (CE) loss as the desired loss, the batch loss would be the sum of the losses of individual samples as follows:

$$L_{\text{batch}} = \sum_{i=1}^{n} L_i = \sum_{i=1}^{n} \sum_{k=1}^{m} -y_k \log a_k, \tag{21}$$

Considering equation 16, the gradient of this loss with respect to the weights can be presented as:

$$\nabla_{\boldsymbol{\theta}} L_{\text{batch}} = \boldsymbol{\rho}_1 \boldsymbol{x}_1^T + \boldsymbol{\rho}_2 \boldsymbol{x}_2^T + \ldots + \boldsymbol{\rho}_n \boldsymbol{x}_n^T. \tag{22}$$

It is important to note that the gradient update is constrained within the subspace spanned by the $n$ input examples.

## F    Singular Value Decomposition (SVD) Explanation

Consider an $m \times n$ matrix $\boldsymbol{R}$, where $m$ is the number of rows and $n$ is the number of columns. The goal of SVD is to factorize matrix $\boldsymbol{R}$ into three separate matrices: $\boldsymbol{U}, \boldsymbol{\Sigma}$, and $\boldsymbol{V}^T$ (transpose of matrix $\boldsymbol{V}$), such that $\boldsymbol{R} = \boldsymbol{U} \boldsymbol{\Sigma} \boldsymbol{V}^T \in \mathbb{R}^{m \times n}$, as presented in figure 5.

$\boldsymbol{U}$: An $m \times m$ orthogonal matrix, where the columns represent the left singular vectors of $\boldsymbol{R}$. $\boldsymbol{\Sigma}$: An $m \times n$ diagonal matrix, where the diagonal entries are the singular values of $\boldsymbol{R}$ (non-negative and

sorted in descending order). $V^T$: An $n \times n$ orthogonal matrix, where the columns represent the right singular vectors of $R$.

Along with singular values and singular vectors, eigen-values and eigen-vectors are also defined. An eigenvalue $\lambda$ and its corresponding eigenvector v of a square matrix $R$ satisfy the equation $R = \lambda V$. Eigen-vectors represent directions in the vector space that are only scaled by the matrix $R$, while eigenvalues represent the scaling factors for those eigen-vectors.

SVD and Relationship to Eigen-values and Eigen-vectors: SVD connects eigen-values and eigen-vectors with the singular values and singular vectors of a matrix. The singular values of $R$ are the square roots of the eigen-values of $RR^T$ or $R^T R$, and the left and right singular vectors are the eigen-vectors of $RR^T$ and $R^T R$, respectively.

Rank and Matrix Approximation: The rank of a matrix $R$ is determined by the number of non-zero singular values in $\Sigma$. By keeping only the largest singular values and their corresponding singular vectors, it is possible to approximate the original matrix $R$ with a lower-rank approximation, which can be useful for dimensionality reduction and noise reduction. We leverage this feature in our approach.

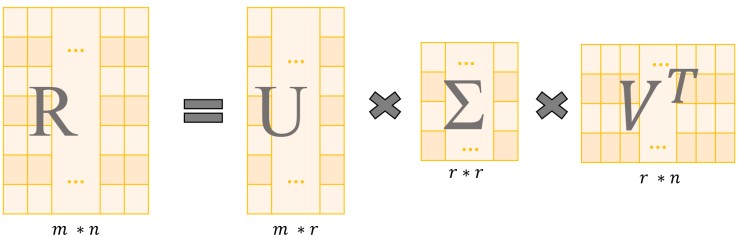

**Figure 5:** Singular Value Decomposition

Properties of SVD:

The singular values in $\Sigma$ are non-negative and arranged in descending order. The columns of $U$ and $V$ are orthonormal, meaning they form an orthogonal basis for their respective vector spaces. The SVD decomposition is unique up to the sign of the singular values and the order of the singular vectors. SVD is a powerful matrix factorization technique that provides a compact representation of a matrix while preserving important structural properties. It finds widespread applications in various fields, such as data analysis, image processing, recommendation systems, and more [10].

# G  Ablation study on the Impact of SVD

In this experiment, our focus is to examine the impact of utilizing singular value decomposition (SVD) in the GradOrth method as an ablation study. To achieve this, we compute the space (as opposed to the subspace) of the pre-trained network using the ID data, without employing SVD. Consequently, the OODness score in GradOrth-NoSVD solely incorporates the orthogonal projection of the new sample onto the *space* of the ID pre-trained network. The experimental results obtained from both the ImageNet and CIFAR benchmarks are presented in tables 12, 13, and 14. The outcomes reported in tables 12, 13, and 14 demonstrate the significant and robust performance of *GradOrth* compared to the GradOrth variant without SVD (*GradOrth-NoSVD*). On the ImageNet benchmark, GradOrth surpasses GradOrth-NoSVD by an average of $14.12\%$ and $10.93\%$ in terms of FPR95 for the ResNet and MobileNet pre-trained networks, respectively. Furthermore, GradOrth exhibits exceptional performance on the CIFAR benchmark, outperforming GradOrth-NoSVD by $6.44\%$ and $8.06\%$ on CIFAR10 and CIFAR100, respectively. The results obtained from this ablation study emphasize the significance of employing SVD in the GradOrth method's OODness score. It underscores the core principle of our approach, which suggests that the essential discriminative features for identifying OOD data reside within the subspace of the ID data.

| Model | Methods | OOD Datasets | | | | | | | | Average | |
|---|---|---|---|---|---|---|---|---|---|---|---|
| | | iNaturalist | | SUN | | Places | | Textures | | | |
| | | FPR95 ↓ | AUROC ↑ | FPR95 ↓ | AUROC ↑ | FPR95 ↓ | AUROC ↑ | FPR95 ↓ | AUROC ↑ | FPR95 ↓ | AUROC ↑ |
| ResNet | GradOrth-NoSVD | 28.31 | 94.30 | 30.09 | 93.73 | 44.18 | 81.43 | 28.17 | 93.21 | 32.69 | 90.67 |
| | GradOrth | 11.04 | 98.00 | 19.61 | 95.76 | 33.67 | 91.78 | 11.19 | 98.06 | 18.57 | 96.31 |
| MobileNet | GradOrth-NoSVD | 38.45 | 90.21 | 38.02 | 89.26 | 49.31 | 90.18 | 28.52 | 89.68 | 38.58 | 89.83 |
| | GradOrth | 26.81 | 93.17 | 30.82 | 93.18 | 40.27 | 89.12 | 12.69 | 97.52 | 27.65 | 93.25 |

**Table 12:** Ablation study to present the importance of SVD in GradOrth. GradOrth presents outstanding performance on average and across all datasets. OOD detection results with **ImageNet-1k** as ID. GradOrth presents outstanding performance on average and across all datasets. The ResNet and MobileNet models are pre-trained solely with ID data from the ImageNet-1k dataset. We use ↑ to denote that larger values are preferable, and ↓ to denote that smaller values are preferable.

| Method | OOD Datasets | | | | | | | | | | | | Average | |
|---|---|---|---|---|---|---|---|---|---|---|---|---|---|---|
| | SVHN | | LSUN-c | | LSUN-r | | iSUN | | Textures | | Places365 | | | |
| | FPR95 ↓ | AUROC ↑ | FPR95 ↓ | AUROC ↑ | FPR95 ↓ | AUROC ↑ | FPR95 ↓ | AUROC ↑ | FPR95 ↓ | AUROC ↑ | FPR95 ↓ | AUROC ↑ | FPR95 ↓ | AUROC ↑ |
| GradOrth-NoSVD | 18.32 | 94.03 | 1.12 | 99.57 | 3.56 | 98.74 | 10.29 | 98.04 | 33.17 | 93.10 | 46.22 | 85.62 | 18.78 | 94.85 |
| GradOrth | 5.84 | 98.72 | 0.81 | 99.78 | 2.33 | 98.71 | 4.25 | 98.32 | 20.63 | 94.77 | 38.22 | 91.64 | 12.34 | 96.99 |

**Table 13:** Ablation study to recognize the importance of SVD in GradOrth. Detailed results on six common OOD benchmark datasets with **CIFAR-10** as ID: Textures [8], SVHN [40], Places365 [63], LSUN-Crop [61], LSUN-Resize [61], and iSUN [58]. For each ID dataset, we use the same DenseNet pre-trained on *CIFAR-10*. ↑ indicates larger values are better and ↓ indicates smaller values are better.

| Method | OOD Datasets | | | | | | | | | | | | Average | |
|---|---|---|---|---|---|---|---|---|---|---|---|---|---|---|
| | SVHN | | LSUN-c | | LSUN-r | | iSUN | | Textures | | Places365 | | | |
| | FPR95 ↓ | AUROC ↑ | FPR95 ↓ | AUROC ↑ | FPR95 ↓ | AUROC ↑ | FPR95 ↓ | AUROC ↑ | FPR95 ↓ | AUROC ↑ | FPR95 ↓ | AUROC ↑ | FPR95 ↓ | AUROC ↑ |
| GradOrth- NoSVD | 31.43 | 93.70 | 9.53 | 96.82 | 57.69 | 87.54 | 51.93 | 92.11 | 42.09 | 90.16 | 55.76 | 82.97 | 41.41 | 90.55 |
| GradOrth | 24.27 | 93.47 | 3.71 | 99.07 | 48.09 | 91.26 | 42.73 | 91.48 | 32.71 | 92.62 | 48.61 | 89.03 | 33.35 | 92.82 |

**Table 14:** Ablation study to recognize the importance of SVD in GradOrth. GradOrth presents outstanding performance on average and across all datasets. Detailed results on six common OOD benchmark datasets with **CIFAR-100** as ID: Textures [8], SVHN [40], Places365 [63], LSUN-Crop [61], LSUN-Resize [61], and iSUN [58]. For each ID dataset, we use the same DenseNet pre-trained on *CIFAR-100*. ↑ indicates larger values are better and ↓ indicates smaller values are better.

# H  *K*-Rank Matrix Approximation

Singular Value Decomposition (SVD) can be used to factorize a rectangular matrix, $\boldsymbol{R} = \boldsymbol{U\Sigma V}^T \in \mathbb{R}^{m \times n}$ into the product of three matrices, where $\boldsymbol{U} \in \mathbb{R}^{m \times m}$ and $\boldsymbol{V} \in \mathbb{R}^{n \times n}$ are orthonomal matrices, and $\boldsymbol{\Sigma}$ is a diagonal matrix that contains the sorted singular values along its main diagonal (Deisenroth et al. [10]). If the rank of the matrix is $r$ ($r \leq \min(m, n)$), $\boldsymbol{R}$ can be expressed as $\boldsymbol{R} = \sum_{i=1}^{r} \sigma_i \boldsymbol{u}_i \boldsymbol{v}_i^T$, where $\boldsymbol{u}_i \in \boldsymbol{U}$ and $\boldsymbol{v}_i \in \boldsymbol{V}$ are left and right singular vectors and $\sigma_i \in diag(\boldsymbol{\Sigma})$ are singular values. $k$-rank approximation of $\boldsymbol{R}$ can be written as, $\boldsymbol{R}_k = \sum_{i=1}^{k} \sigma_i \boldsymbol{u}_i \boldsymbol{v}_i^T$, where $k \leq r$ and its value can be chosen by the smallest $k$ that satisfies the norm-based criteria : $||\boldsymbol{R}_k||_F^2 \geq \epsilon_{th}||\boldsymbol{R}||_F^2$. Here, $||.||_F$ denotes the Frobenius norm of the matrix and $\epsilon_{th} \in (0, 1)$ is the threshold hyperparameter.

$$\boldsymbol{R}_k = \sum_{i=1}^{k} \sigma_i \boldsymbol{u}_i \boldsymbol{v}_i^T \tag{23}$$

$$||\boldsymbol{R} - \boldsymbol{R}_k||^2 = \sum_{i=1}^{m} \sum_{j=1}^{n} |a_{ij} - \hat{a}_{ij}|^2 \sum_{j=k+1}^{r} \sigma_j^2 \quad 0 \leq k \leq n \text{ where } R_k = \hat{a}_{ij}. \tag{24}$$

The degree to which $R_k$ approximates $R$ depends on the sum of the r-k smallest singular values squared. As k approaches r, this sum becomes progressively smaller and eventually goes to zero at $k = r$. To provide a convenient measure for this behavior independent of the size of $R$, let us consider the normalized matrix approximation ratio

$$\epsilon_{th}(k) = \frac{||R_k||}{||R||} = [\frac{\sigma_1^2 + \sigma_2^2 + ... + \sigma_k^2}{\sigma_1^2 + \sigma_2^2 + ... + \sigma_r^2}]^{\frac{1}{2}}, 1 \leq k \leq r. \tag{25}$$

Clearly, this normalized ratio approaches its maximum value of 1 as k approaches r. For matrices of low effective rank, $\epsilon_{th}(k)$ is close to 1 for values of k significantly smaller than r. On the other hand, matrices for which $m$ must take on high values (i.e., $k \approx r$) to achieve a $\epsilon_{th}(k)$ near 1 are said to be of high effective rank [3].

# I  Impact of the Threshold Parameter ($\epsilon_{th}$) on GradOrth Performance

The hyperparameter $\epsilon_{th}$, confined to the range $(0, 1)$, serves as a threshold that influences the selection of the value of $k$ in the matrix $k$-rank approximation. The initial $k$ column vectors within matrix $\boldsymbol{U}$

encompass the most pivotal input (representation) space for the pre-trained network. We conducted an experiment to assess the impact of $\epsilon_{th}$ on GradOrth's performance, with results outlined in table 15. Notably, values of $\epsilon_{th}$ near 1 exhibit substantial effectiveness, and we empirically set it to $0.97$.

| Model | OOD Datasets | | | | | | | | | | | | | |
|---|---|---|---|---|---|---|---|---|---|---|---|---|---|---|
| | SVHN | | LSUN-c | | LSUN-r | | iSUN | | Textures | | Places365 | | Average | |
| | FPR95 ↓ | AUROC ↑ | FPR95 ↓ | AUROC ↑ | FPR95 ↓ | AUROC ↑ | FPR95 ↓ | AUROC ↑ | FPR95 ↓ | AUROC ↑ | FPR95 ↓ | AUROC ↑ | FPR95 ↓ | AUROC ↑ |
| GradOrth-$\epsilon_{th} = 0.80$ | 24.57 | 93.39 | 3.82 | 98.87 | 48.19 | 91.15 | 42.80 | 91.34 | 32.83 | 92.51 | 48.70 | 88.93 | 33.49 | 92.70 |
| GradOrth-$\epsilon_{th} = 0.90$ | 24.36 | 93.42 | 3.75 | 99.00 | 48.12 | 91.24 | 42.74 | 91.46 | 32.76 | 92.58 | 48.64 | 89.01 | 33.39 | 92.78 |
| GradOrth-$\epsilon_{th} = 0.97$ | 24.27 | 93.47 | 3.71 | 99.07 | 48.09 | 91.26 | 42.73 | 91.48 | 32.71 | 92.62 | 48.61 | 89.03 | 33.35 | 92.82 |

**Table 15:** Detailed results on six common OOD benchmark datasets considering different values for $\epsilon_{th}$ used in $k$-rank matrix approximation. For the pre-trained network and ID dataset, we use DenseNet pre-trained on *CIFAR-100*.

# J  Details of Experiments

## J.1  Model and Hyper Parameter

In our empirical studies and experiments, we adopt an experimental setting that aligns with the state-of-the-art (SOTA) approaches, specifically ASH [12] and DICE [47] on the CIFAR dataset, as well as ReAct [48] on the ImageNet dataset. The datasets and model architectures utilized in our experiments are summarized in table 16.

For the CIFAR-10 and CIFAR-100 experiments, we employ the six OOD datasets employed in the DICE study [47]: SVHN [40], LSUN-Crop [61], LSUN-Resize [61], iSUN [58], Places365 [63], and Textures [8]. The ID dataset used in these experiments corresponds to the respective CIFAR dataset. The model architecture employed is a pre-trained DenseNet-101 [21].

For our ImageNet experiments, we adhere to the precise setup as outlined in the ReAct study [48] and [12]. The ID dataset employed in this context is ImageNet-1k, while the OOD datasets consist of iNaturalist [53], SUN [56], Places365 [63], and Textures [8]. The network architectures utilized in these experiments are ResNet50 [16] and MobileNetV2 [44]. All networks undergo pre-training using the ID data and remain unaltered post-training, with their parameters remaining unchanged during the OOD detection phase. The performance of the baselines primarily relies on ASH and VRA. When conducting experiments involving gradient-based methods such as GradNorm [23] and ExGrad [24], we re-run the experiments using the code provided by the respective authors, as there may be variations between our pre-trained network over ID data and the models employed by the authors themselves.

| ID Dataset | OOD Datasets | Model architectures |
|---|---|---|
| CIFAR-10 | SVHN, LSUN C, LSUN R, iSUN, Places365, Textures | DenseNet-101 |
| CIFAR-100 | SVHN, LSUN C, LSUN R, iSUN, Places365, Textures | DenseNet-101 |
| ImageNet | iNaturalist, SUN, Places365, Textures | ResNet50, MobileNetV2 |

**Table 16:** The datasets and models we used in our OOD experiments range from moderate to large scale, including evaluations of up to 10 OOD datasets and three architectures.

## J.2  Datasets

**ImageNet Benchmark, Large-scale evaluation**   In this study, the ImageNet-1k dataset [11] is employed as the ID dataset. The evaluation of the proposed approach is conducted on four OOD test datasets, following the experimental setup outlined in [23]:

- **iNaturalist**  The dataset introduced by [53], referred to as "iNaturalist", comprises a substantial collection of 859,000 images featuring various plant and animal species. These images span more than 5,000 distinct species. To facilitate efficient processing, each image within the dataset is resized to ensure that the maximum dimension does not exceed 800 pixels. For the evaluation phase, a subset of 10,000 images is randomly sampled from a set of 110 classes. Importantly, these selected classes are disjoint from the ImageNet-1k dataset, thereby ensuring the validity and independence of the evaluation process.
- **SUN**  that is introduced by Xiao et al., encompasses a vast collection of over 130,000 images representing various scenes. These scenes are categorized into 397 distinct categories.

Notably, it is important to acknowledge that there are overlapping categories between the SUN dataset and the ImageNet-1k dataset. For the evaluation process, a subset of 10,000 images is randomly sampled from a set of 50 classes. Importantly, these selected classes are disjoint from the labels present in the ImageNet dataset. This ensures the integrity and independence of the evaluation conducted in the study.

- **Places** The dataset introduced by [63], commonly referred to as "Places" in the research literature, is another notable scene dataset that exhibits similar concept coverage as the SUN dataset. In this study, a carefully selected subset of 10,000 images is utilized from a total of 50 classes. Importantly, these selected classes are intentionally excluded from the ImageNet-1k dataset, ensuring that the evaluation process remains independent and free from any potential overlap with the aforementioned dataset.

- **Textures** [7] consisting of 5,640 real-world texture images categorized into 47 distinct categories, is utilized in this research study. For the purpose of evaluation, the entire dataset is utilized, ensuring comprehensive coverage across all available categories.

**CIFAR Benchmark** The CIFAR-10 and CIFAR-100 datasets, introduced by [26], are extensively employed as ID datasets in the existing literature. CIFAR-10 comprises 10 classes, while CIFAR-100 consists of 100 classes. In line with standard practices, the dataset split utilized in this study consists of 50,000 training images and 10,000 test images. To evaluate the proposed approach, four commonly employed OOD datasets are utilized. The specific OOD datasets used in this evaluation are listed below:

- **SVHN** [40] The Street View House Numbers (SVHN) dataset consists of color images depicting house numbers. The dataset encompasses ten distinct classes corresponding to the digits 0-9. In this research study, the entire test set comprising 26,032 images is employed for evaluation purposes.

- **LSUN** [61] includes a collection of 10,000 testing images featuring 10 different scenes. In this research study, image patches of size 32x32 are randomly cropped from the LSUN dataset to facilitate analysis and experimentation.

- **Places365** [63] comprises a vast collection of large-scale photographs depicting scenes across 365 distinct scene categories. Notably, the test set of this dataset consists of 900 images per category. For the evaluation phase, a random sample of 10,000 images is drawn from the test set, thereby ensuring a representative subset for rigorous analysis and assessment.

- **Textures** [7] includes a comprehensive collection of 5,640 real-world texture images, classified into 47 distinct categories. In this research study, the entire dataset is utilized for evaluation, enabling a thorough examination of the performance and capabilities of the proposed approach across all available texture categories.

### J.3  Baselines

For the reader's convenience, we summarize in detail a few common techniques for defining OOD scores that measure the degree of ID-ness on the given sample. By convention, a higher (lower) score is indicative of being in-distribution (out-of-distribution).

**MSP [18]** utilizes probabilities obtained from softmax distributions to distinguish between correctly classified and erroneous or out-of-distribution examples. The baseline method relies on the observation that correctly classified examples tend to have higher maximum softmax probabilities than misclassified and out-of-distribution examples.

**ODIN [32]** is based on the observation that using temperature scaling and adding small perturbations to the input can separate the softmax score distributions between in- and out-of-distribution images, allowing for more effective detection.

**Mahalanobis [29]** utilizes multivariate Gaussian distributions to effectively model the class-conditional distributions of softmax neural classifiers. Additionally, they employed Mahalanobis distance-based scores as a means of detecting out-of-distribution samples.

**Energy Score [34]**  The concept of utilizing energy scores for estimating out-of-distribution uncertainty was initially introduced by Liu et al.. The energy function employed in their approach maps the logit outputs to a scalar value denoted as $S_{\text{Energy}}(x; f) \in \mathbb{R}$. Notably, this scalar value tends to be relatively lower for in-distribution data. It is important to mention that the authors adopted the convention of using the negative energy score for OOD detection, ensuring that $S(x; f)$ exhibits higher values for ID data and lower values for OOD data.

**GradNorm [23]**  computes the norm of the gradients of a neural network with respect to an input. The norm of the gradients is a measure of how sensitive the network is to the input. Inputs with high norms are more likely to be OOD because they are more likely to cause the network to make a mistake.

**Exgrad[24]**  calculates the expected norm of the gradients of a neural network with respect to an input. The expected norm of the gradients is a measure of how sensitive the network is to the input.

**DICE [47]**  aims at selectively utilizing a subset of significant weights to compute the output for OOD detection. By employing the technique of sparsification, the network effectively avoids incorporating irrelevant information into the output, thereby enhancing its OOD detection capabilities.

**ReAct [49]**  This approach is based on the observation that OOD data trigger distinctive activation patterns in neural networks. ReAct selectively rectifies and truncates the activations of specific hidden units, reducing overconfident predictions on OOD data.

**Variational Rectified Activation (VRA) [57]**  VRA leverages the variational method to find the optimal operation for maximizing the gap between ID and OOD data. It introduces suppression and amplification operations for abnormally low, high, and intermediate activations, unlike ReAct which only focuses on high activations. VRA uses piecewise functions to simulate these operations.

**ASH [12]**  This method removes a large portion of activations and adjusts the remaining ones. The remaining activations (e.g., 10%) are either simplified or lightly adjusted. The simplified activation representation is then propagated through the rest of the network to generate scores for both classification and OOD detection. The energy score, calculated from the logits, is commonly used for OOD detection, although the softmax score can also be used.

## J.4  Software and Hardware

**Software**  We run all experiments with Python 3.8.0 and PyTorch 1.12.1.

**Hardware**  All experiments are run on NVIDIA RTX 3090.

