# OpenReview forum: "GradOrth: A Simple yet Efficient Out-of-Distribution Detection with Orthogonal Projection of Gradients"
_NeurIPS.cc/2023/Conference — NeurIPS 2023 poster_

### Official Review · Reviewer_MCXS · 2023-07-03

**Soundness:** 3 good
**Presentation:** 3 good
**Contribution:** 3 good
**Rating:** 6
**Confidence:** 4

**Summary:**

The paper proposes to project the per-example parameter-gradients of the prediction head onto the orthogonal subspace of the learned features and uses the projection norm as the OODness score. The key motivation for doing so is that the parameter gradients lie in the subspace of the corresponding input features. Key highlights of this method are that it is a post-hoc detection method and is computationally efficient requiring few in-distribution examples for computing the orthogonal subspace of the features. It achieves good detection scores.

**Strengths:**

The main strength of the paper is in the idea building upon the observation that stochastic-gradient descent updates lie in the subspace of input features for detecting OOD examples. The paper is very well written and the reports various ablation studies to clearly illustrate the effects of the hyperparameter design-choices.

**Weaknesses:**

While the empirical results show good performance on the commonly considered benchmarks, I believe that it is essential to incorporate results related to Semantic Outliers [1] and Near-distribution Outliers, such as the comparison between CIFAR10 and CIFAR100 [2, 3]. Section 4.3 in [4] can serve as an additional reference for conducting a similar evaluation. It is worth noting that it is okay if the results are not very strong; rather, the key objective is to present a comprehensive assessment of the estimator and update the limitations section accordingly. Furthermore, it would be beneficial to include a clear rationale behind the selection of baseline algorithms -- considering a comparison with [3] would be appropriate if the primary empirical contribution focuses on far-OOD detection.

I also thought that a qualitative analysis of the estimator could also be helpful in illustrating the cases where GradOrth works best and its failure cases as compared to other baseline estimators.

[1] Ahmed, F., & Courville, A. (2020). Detecting Semantic Anomalies. Proceedings of the AAAI Conference on Artificial Intelligence, 34(04), 3154-3162.

[2] Fort, S., Ren, J., & Lakshminarayanan, B. (2021). Exploring the limits of out-of-distribution detection. Advances in Neural Information Processing Systems, 34, 7068-7081.

[3] Sastry, C. S., & Oore, S. (2020, November). Detecting out-of-distribution examples with gram matrices. In International Conference on Machine Learning (pp. 8491-8501). PMLR.

[4] Hsu, Yen-Chang, Yilin Shen, Hongxia Jin, and Zsolt Kira. "Generalized odin: Detecting out-of-distribution image without learning from out-of-distribution data." In Proceedings of the IEEE/CVF Conference on Computer Vision and Pattern Recognition, pp. 10951-10960. 2020.

**Questions:**

1. I believe that this work has some connections with [1, 2] and can be used to complement the abilities of GradOrth in interesting ways.
2. In Appendix B, you show that the gradients obtained from all layers perform a bit worse than the final layer alone: do you think that using a different scaling factor for each layer can help in building a better estimator than just the last layer?
3. The algorithm requires computing per-example gradients of the loss wrt parameters: how did you do this for processing a batch of examples while staying as computationally efficient as possible? A short discussion about this would be useful to include in the paper.
4. Apart from the obvious typos in the equation on line 124 of subsection 2.2, could you please elaborate why you compare the frobenius norms of matrices ${\bf R}$ and ${\bf R}_k$ instead of the obvious choice of using the difference $|| {\bf R}_k - {\bf R}||_F$  for identifying the appropriate $k$? Is this related to the inequality between frobenius norms and squared sum of eigenvalues -- if so, it would be good to explain this.
5. In section (3), it would be good to include the shapes of matrices wherever relevant.
6. If you have any preliminary results/thoughts about the weaknesses section, please include it in the replies.

(The new experimental results highlight the effectiveness of the method and I raise my score.)

[1] Koh, Pang Wei, and Percy Liang. "Understanding black-box predictions via influence functions." ICML 2017.

[2] Madras, David, James Atwood, and Alex D'Amour. "Detecting underspecification with local ensembles." ICLR 2020.

**Limitations:**

Yes, the limitations are adequately addressed.

---

> ### Author Rebuttal · Authors · 2023-08-10
>
>
> Dear Reviewer MCXS,
>
> Thanks for your thorough evaluation of our paper. Your suggestions will significantly enhance the comprehensiveness and clarity of our work. We would like to address your comments and queries as follows:
>
> - Experiments on near-OOD and semantic outliers:
>
> Thank you for your valuable suggestion! Our empirical findings have consistently demonstrated remarkable and reliable performance across both near-ID OOD detection and semantic outlier scenarios.
>
> In our comprehensive near-ID experiments, outlined in Table 1 of the attached PDF, our approach showcased notably robust results, particularly evident within the CIFAR datasets. Furthermore, when applied to the SVHN dataset, our GradOrth approach displayed competitive performance, securing a position among the top three methods. These experiments were executed within the experimental parameters specified in [5].
>
> Furthermore, our GradOrth method has consistently displayed exceptional and unwavering performance in evaluating semantic outliers. As depicted in Table [2], GradOrth outperforms other baseline techniques in handling semantic shifts (S), a notably challenging OOD task, particularly on the real-B dataset. It is worth noting that the performance of GradOrth rivals that of the Mahalanobis method while maintaining a lower computational complexity.
>
> - using [1, 2] in Gradorth:
>
> Thanks for your valuable suggestion. In our revision, we would include citations to the studies and plan to consider them for our future research.
>
> -  Scaling factor for each layer
>
>  Our comprehensive empirical analyses reveal that the incorporation of layer-wise scaling factors does not lead to a significant enhancement.  The foremost challenge tied to the integration of layer-wise scaling revolves around precisely ascertaining suitable scaling factors for each layer. Importantly, the preference for the last layer emerges due to its potential to curtail time complexity, rendering it a more appealing option.
>
>
> - Per-sample gradient computations:
>
> Per-sample gradient computations can be efficiently performed using available libraries and functions. For PyTorch, the official website provides details on utilizing the transforms function for this purpose.  Similarly, in TensorFlow, efficient per-sample gradient computation can be achieved through the vectorized-map function, as elaborated on the official website.
>
> Moreover, several researchers have contributed to the development of efficient per-sample gradient computation methods, which are extensively discussed in their respective papers and accompanied by code repositories on GitHub, such as:
>
>    - Rochette, Gaspar, Andre Manoel, and Eric W. Tramel. "Efficient per-example gradient computations in convolutional neural networks."
>
>    - Goodfellow, Ian. "Efficient per-example gradient computations."
>
>
> -  could you please elaborate why you compare the frobenius norms of matrices and instead of the obvious choice of using the difference for identifying the appropriate ?
>
> Thanks for pointing out our typo, we would fix it in our revision.
>
> We follow the theorem presented in [3] for $k$-rank approximation. It is a convenient measure for $k$-rank approximation independent of the size of $R$.
> The detailed explanation is as follow:
> Singular Value Decomposition (SVD) can be used to factorize a rectangular matrix, $A =U \Sigma V^T \in \mathbb{R}^{m\times n}$ into the product of three matrices, where $U\in \mathbb{R}^{m\times m}$ and $V\in \mathbb{R}^{n\times n}$ are orthonomal matrices, and $\Sigma $ is a diagonal matrix that contains the sorted singular values along its main diagonal [4]. If the rank of the matrix is $r$ ($r\leq \text{min}(m,n)$), $R$ can be expressed as $R=\sum_{i=1}^r \sigma_i u_i v_i^T$, where $ u_i \in U $  and  $ v_i \in V $ are left and right singular vectors and $\sigma_i \in diag({\Sigma})$ are singular values. $k$-rank approximation of $R$ can be written as,
> $ R_k=\sum_{i=1}^k \sigma_i u_i v_i^T$, where $k\leq r$ and its value can be chosen by the smallest $k$ that satisfies the norm-based criteria : $\lvert\lvert  R_k \rvert\rvert_F^2 \geq \epsilon_{th}\lvert\lvert R\rvert\rvert_F^2$. Here, $\lvert\lvert.\rvert\rvert_F$ denotes the Frobenius norm of the matrix and $\epsilon_{th} \in(0,1)$ is the threshold hyperparameter.
>
> $R_k=\sum_{i=1}^k \sigma_i u_i  v_i^T$
>
> $\lvert\lvert  R - R_k \rvert \rvert ^2 = \sum_{i=1}^m \sum_{j=1}^n \lvert a_{ij} - \hat{a}_{ij} \rvert ^2  \sum_z^r \sigma_z^2  $ , $ 0 \leq k \leq n$
>
> such that  $  R_k=\hat{a}_{ij}$ , $z = k+1 $
>
> The degree to which $R_k$ approximates $R$ depends on the sum of the r-k smallest singular values squared. As k approaches r, this sum becomes progressively smaller and eventually goes to zero at $k = r$. To provide a convenient measure for this behavior independent of the size of $R$, let us consider the normalized matrix approximation ratio
>
> $\epsilon_{th}(k)=\frac{\lvert \lvert R_k \rvert \rvert}{\lvert \lvert R \rvert \rvert} = [\frac{\sigma_1^2+\sigma_2^2+ ... + \sigma_k^2}{\sigma_1^2+\sigma_2^2+ ... + \sigma_r^2}]^{\frac{1}{2}}, 1\leq k \leq r$.
>
> Clearly, this normalized ratio approaches its maximum value of 1 as $k$ approaches $r$. For matrices of low effective rank, $\epsilon_{th}(k)$ is close to 1 for values of $k$ significantly smaller than $r$. On the other hand, matrices for which $m$ must take on high values (i.e., $k \approx r$) to achieve a $\epsilon_{th}(k)$ near 1 are said to be of high effective rank [3] .
>
> [1] Pang Wei Koh and Percy Liang. Understanding black-box predictions via influence functions.PMLR, 2017.
>
> [2] David Madras, James Atwood,  Alex D’Amour. Detecting underspecification with local ensembles.
>
> [3] JAMES A. CADZOW. Chapter 9 - Spectral Analysis. San Diego, 1987.
>
> [4] Marc Peter Deisenroth, A. Aldo Faisal, and Cheng Ong. Mathematics for Machine Learning.
>
> [5] Chandramouli  Sastry and Sageev Oore. Detecting out-of-distribution examples with gram matrices. PMLR, 2020

---

> > ### Comment · Reviewer_MCXS · 2023-08-13
> > **Response to Rebuttal.**
> >
> > Thank you for your extra experiments and answers to my questions.
> > I think that it will be good to add some of this discussion into the paper.

---

> > > ### Author Response · Authors · 2023-08-15
> > >
> > > Dear Reviewer MCXS,
> > >
> > > Thanks again for reviewing our work. We do appreciate your comments/ suggestions and integrate them in our revision.

---

### Official Review · Reviewer_6mNB · 2023-07-04

**Soundness:** 3 good
**Presentation:** 4 excellent
**Contribution:** 2 fair
**Rating:** 6
**Confidence:** 4

**Summary:**

The work proposes GradOrth, a simple post-hoc OOD detection method that seeks to improve on existing gradient-based OOD detection approaches by leveraging a selected lower-rank subspace of the gradient most relevant for OOD detection. The resulting GradOrth method shows superior OOD detection performance on a wide range of OOD detection tasks.

**Strengths:**

1. The paper is clearly written and investigates a generally underexplored aspect of OOD detection.
2. The resulting GradOrth post-hoc method is easy to implement and shows strong empirical performance on a wide range of OOD detection tasks.
3. The work presents a series of relevant ablation studies ranging from the choice of norm to the number of ID samples used to compute the subspace.

**Weaknesses:**

1. Some comparable and competitive methods are missing. For example, the reviewer would like to see empirical results comparing DICE [1] with GradNorm [2] and KNN [3].
2. Additionally, the reviewer would like to see empirical result deviations, as randomness with respect to the chosen ID samples, used to compute the subspace, may be relevant to the empirical performance of GradOrth.

[1] Yiyou Sun and Yixuan Li. Dice: Leveraging sparsification for out-of-distribution detection. In Proceedings of European Conference on Computer Vision, 2022.

[2] Rui Huang, Andrew Geng, and Yixuan Li. On the importance of gradients for detecting distributional shifts in the wild. In Advances in Neural Information Processing Systems, 2021.

[3] Yiyou Sun, Yifei Ming, Xiaojin Zhu, and Yixuan Li. Out-of-distribution detection with deep nearest neighbors. In International Conference on Machine Learning, 2022.

**Questions:**

1. Is there a reason why DICE's sparsification can not be implemented alongside a method like GradNorm?
2. Previous attempts at Gradient-based approaches have shown L1 norm to be superior, do the authors have some intuition on the reasoning why L2 norm is stronger for GradOrth?

**Limitations:**

The authors did not directly address societal impacts however there should be no negative impacts from this work.

---

> ### Author Rebuttal · Authors · 2023-08-10
>
>
> Dear Reviewer 6mNB,
>
> We  appreciate your thoughtful evaluation of our work. Your feedback provides valuable insights that will undoubtedly contribute to the refinement and enhancement of our research. We would like to address your comments and concerns as follows:
>
> We have compared our method against DICE [1] and the results are presented in tables 1, 2, and 3 of the paper. Thanks for pointing out KNN [3] method, we would add KNN as another baseline in our experiments and cite in our paper. It is notable that GradOrth presents outstanding performance in competence with KNN.
>
> - Experiments on DICE+ GradNorm
>
> Thanks for this interesting suggestion!
> We leverage DICE [1] network architecture provided in authors Github and plug it in GradNorm [2] to get an implementation of DICE+ GradNorm. We run experiment on Resnet50 pretrained on ImageNet as ID and tested it on 4 OOD datasets: iNaturalist, SUN, Places, and Textures. Our experimental studies did not present any improvement using the scarification threshold at 90%.
>
>
> - Previous attempts at Gradient-based approaches have shown L1 norm to be superior, do the authors have some intuition on the reasoning why L2 norm is stronger for GradOrth?
>
> While both GradNorm and GradOrth leverage gradients for OOD detection, they adopt differing methodologies. GradNorm places a strong emphasis on the L1 norm, utilizing it to assess gradient magnitudes and discern disparities between in-distribution (ID) and OOD data. This focus on gradient magnitude forms the crux of GradNorm's approach.
>
> In contrast, GradOrth takes a unique path by employing orthogonal gradient projection within a pre-trained network subspace. In this context, the L2 norm is more suitable for approximating projection length, offering a distinct perspective from the L1 norm employed by GradNorm.
>
> - Additionally, the reviewer would like to see empirical result deviations, as randomness with respect to the chosen ID samples, used to compute the subspace, may be relevant to the empirical performance of GradOrth.
>
> We appreciate your highlighting the concern regarding our randomness computations. To ensure a comprehensive and equitable comparison, we incorporate two levels of randomness: trial-level and sample-level.
>
> For our experimental investigations, we adopt an average-results-over-5-runs approach. In each run, distinct random seeds are employed to select random samples from each class, generating small subsets of in-distribution data. Subsequently, we compute the subspace of the pre-trained network based on these subsets. The OOD scores of the test data are then calculated, and FPR95 and AUROC scores are derived. This process is repeated five times, and the average of these five runs is reported as the final score. As documented in our paper, each subspace is computed using ten random samples per class in the ImageNet benchmark and five random samples per class in CIFAR benchmarks.
>
> In response to your feedback, we have taken into account the standard deviations estimated across these five runs. These deviations are presented in Tables 5, 6, and 7 within the attached PDF file, and we incorporate this information in our revised version.
>
> [1] Yiyou Sun and Yixuan Li. Dice: Leveraging sparsification for out-of-distribution detection. In Proceedings of European Conference on Computer Vision, 2022.
>
> [2] Rui Huang, Andrew Geng, and Yixuan Li. On the importance of gradients for detecting distributional shifts in the wild. In Advances in Neural Information Processing Systems, 2021.
>
> [3] Yiyou Sun, Yifei Ming, Xiaojin Zhu, and Yixuan Li. Out-of-distribution detection with deep nearest neighbors. In International Conference on Machine Learning, 2022.

---

> ### Comment · Reviewer_6mNB · 2023-08-14
>
> Thank you to the authors for the detailed response to the reviewer's questions. I would like to encourage the authors to incorporate some of these responses into the final draft. Beyond that, I have no further questions and have adjusted my rating accordingly.

---

> > ### Author Response · Authors · 2023-08-15
> >
> >
> > Dear Reviewer 6mNB,
> >
> > Thank you once more for reviewing our work. Your feedback is greatly valued and we diligently incorporate your suggestions into our revision.

---

### Official Review · Reviewer_81Lc · 2023-07-06

**Soundness:** 3 good
**Presentation:** 4 excellent
**Contribution:** 3 good
**Rating:** 7
**Confidence:** 4

**Summary:**

The paper proposes an OOD detection method by exploiting the orthogonal projections of gradients to a subspace. Once a model is trained on an in-distribution (ID) dataset, the k most significant features in the last layer of a network are found using SVD using a small subset of n ID samples. Then, the gradient of the loss given an input data w.r.t. the parameters of the last layer are projected to this significant subspace. Finally, the norm of this projection is used as OOD score which is expected to be large of OOD samples and low for the ID ones. The experiments are performed on 3 ID datasets (ImageNet, CIFAR10 and CIFAR100) with different architectures and OOD datasets. The results demonstrate that the proposed method achieves the highest average scores across all OOD datasets for each ID dataset.

**Strengths:**

- The idea of using projection of gradients for OOD detection is interesting and, to the best of my knowledge, is novel.
- The paper presents the idea quite clearly and supports many of the claims (there is one exception I will mention in the weaknesses section) with sufficient experimental results.
- The paper acknowledges the fact the performance of many OOD methods varies a lot in different datasets and mentions that the performance of the method holds for the datasets used in the paper.

**Weaknesses:**

- The idea presented in the paper is interesting and the paper presents sufficient experiments on the common OOD benchmarks in the literature. However, the results of almost all methods are already very close to each other in this benchmark and there is no clear winner, as also mentioned in the paper. Some baseline methods may perform slightly better than those reported in the paper with a more careful hyperparameter tuning. For example, the original Mahalanobis paper reports AUROC of 99.2 on LSUN dataset when CIFAR10 is in-distribution while the paper reports 97.09 in Table 2.
Therefore, OOD detection methods should be evaluated on more realistic and challenging scenarios to better assess their performance, such as in near-OOD detection settings as in [1] where CIFAR10 is in-distribution data and CIFAR100 is OOD and vice versa. The paper lacks an evaluation of the methods in such a more realistic and challenging setting where most OOD detection methods fail. Even if the results in this setting are not good, this should be presented as a limitation of the work.

[1] Lee et al. A Simple Unified Framework for Detecting Out-of-Distribution Samples and Adversarial Attacks

- The paper claims that the method has a lower computational complexity than existing methods such as Mahalanobis. However, this claim is not supported by quantitative results. As the paper claims, extracting features at intermediate layers in Mahalanobis may not bring significant computational complexity.

**Questions:**

- Given the similar performance of all OOD methods on this benchmark, why should one use the proposed method instead of the others?
- How does the method perform in a near-OOD setting?
- Is the computational time of the proposed method significantly lower than the other methods, as claimed in the paper? Please provide quantitative results.


**Limitations:**

The paper discusses some limitations of the evaluation of OOD methods in general rather than the limitations of this method.

I don't see any potential negative societal impact of this work!

---

> ### Author Rebuttal · Authors · 2023-08-09
>
>
> Dear Reviewer 81LC,
>
> We greatly appreciate your insightful review of our paper. Your comprehensive assessment of our methodology and experimental results provides valuable feedback that will undoubtedly contribute to the refinement and advancement of our work. Please find the responses to your questions in the following:
>
> - Given the similar performance of all OOD methods on this benchmark, why should one use the proposed method instead of the others?
>
>      - "Reliability and stability" in "Far-OOD" detection: Expounded upon in section 4 of our paper, GradOrth consistently outperforms a majority of state-of-the-art methods across diverse benchmarks, affirming its "reliability and stability" in "far-OOD" detection tasks spanning various datasets and network architectures. This remarkable consistency and superior performance distinguish GradOrth from its counterparts.
>
>      - "Reliability and stability" in "Near-OOD" detection: As evidenced by the data presented in tables 1 and 2 within the attached PDF file, GradOrth's "reliability and stability" are showcased in "both near-OOD detection and more challenging tasks", spanning a diverse range of datasets and network architectures.
>
>      - "Efficient and Simple: GradOrth's time complexity, akin to GradNorm, delivers heightened performance while maintaining a lower computational burden compared to the Mahalanobis method. This advantageous time complexity, particularly when applied to large networks, positions GradOrth as a favorable choice.
>
>
>
> - How does the method perform in a near-OOD setting?
>
> Thank you for your valuable suggestion in conducting further experimental investigations pertaining to near-ID OOD detection. Our empirical findings have consistently demonstrated remarkable and reliable performance across both near-ID OOD detection and semantic outlier scenarios.
>
> In our comprehensive near-ID experiments, outlined in Table 1 of the attached PDF, our approach showcased notably robust results, particularly evident within the CIFAR datasets. Furthermore, when applied to the SVHN dataset, our GradOrth approach displayed competitive performance, securing a position among the top three methods. These meticulous experiments were executed within the experimental parameters specified in [2].
>
> Furthermore, our GradOrth method has consistently displayed exceptional and unwavering performance in evaluating semantic outliers. As depicted in Table 2 in  the attached PDF, GradOrth outperforms other baseline techniques in handling semantic shifts (S), a notably challenging OOD task, particularly on the real-B dataset. It is worth noting that the performance of GradOrth rivals that of the Mahalanobis method while maintaining a lower computational complexity.
>
>
> - Is the computational time of the proposed method significantly lower than the other methods, as claimed in the paper?
>
> GradOrth's time complexity is akin to that of GradNorm [1], except for the additional pre-processing step to compute the pre-trained network subspace. As it is explained in GradNorm, Mahalanobis requires collecting feature representations from
> intermediate layers over the entire training set, which is expensive for large-scale datasets such as
> ImageNet. In contrast, GradOrth can be conveniently used through a simple gradient projection without hyper-parameter tuning or additional training.
>
> [1] Rui Huang, Andrew Geng, and Yixuan Li. On the importance of gradients for detecting distributional shifts in the wild. Advances in Neural Information Processing Systems, 34:677–689, 2021.
>
> [2] Chandramouli Shama Sastry and Sageev Oore. Detecting out-of-distribution examples with gram matrices. In International Conference on Machine Learning, pages 8491–8501. PMLR, 2020.

---

> > ### Comment · Reviewer_81Lc · 2023-08-13
> >
> > Thanks to the authors for the rebuttal addressing my comments. I have no additional comments for this paper and support its acceptance. I will adjust my rating accordingly.

---

> > > ### Author Response · Authors · 2023-08-15
> > >
> > > Dear Reviewer 81LC,
> > >
> > > We appreciate your review of our work and value your feedback. Your comments and suggestions are important to us, and we incorporate them into our revision.

---

### Official Review · Reviewer_fiAZ · 2023-07-07

**Soundness:** 2 fair
**Presentation:** 1 poor
**Contribution:** 2 fair
**Rating:** 5
**Confidence:** 5

**Summary:**

This paper focuses on OOD detection based on gradient information. It seems based on the framework of GradNorm (the paper is confusing on this) to use the gradient response as the score for detecting OOD data. Specifically, it uses SVD based subspace modeling and projection of the testing sample’s gradient as the OOD detection score. As shown in the reported results, the proposed method performs well and achieves better performance on many measurements compared to the previous methods.

**Strengths:**

- Generally, it is well-motivated to use subspace analysis to enhance the gradient-based OOD detection approach.
- Experiments are conducted comprehensively on different benchmark datasets and with different backbone models.

**Weaknesses:**

- The formulation of the approach is not clearly introduced or missed, which is the basis of the work. This makes the paper very confusing and influences the understanding of the model’s behavior. The details are in the following:
     - How is gradient calculated on the testing samples when only input images are available in inference time? Does the paper use the model formulation similar to GradNorm (or ExGrad), which uses the gradient from the loss by letting the model predict uniform probability in testing? Or the proposed model uses some different strategies maybe related to the formulations in sec 2.1 (seems not make sense)? No matter what, the authors may provide a rigorous formulation and discuss the technical motivations scientifically.
    - This influences understanding and evaluating the specific motivation and technical soundness of the subspace modeling. Is the modeling focused on the subspace of representation, gradients, or the relationship of them? Since the subspace $S^L$ obtained in Algo 1 is for the representation, understanding how the gradient is calculated in Algo 2 is important.

- The evaluations are with some problems, with details as the following points:
    - There are several operations involving randomness. I cannot find whether the paper evaluate how the randomness may influence the performance and may report the mean and std of the results while given different random seeds.
    - Specifically, the ID representations and SVD-based subspace models are from selected samples. The results should be sensitive to the quality of the selected ID samples, especially when the number is small. I can see experiments studying how the number of selected samples may influence the results. How the randomness may influence the selection and then the results?
    - The authors may consider analyzing and evaluating the distribution of the scores of ID and OOD samples, as in previous works.


---after rebuttal---
The proposed method does make sense and performs slightly better than the main competitors, without supersize. It is a combination of several existing methods, such as GradNorm and the projection-based methods of CL. I didn't mention this issue since the representation in the original paper is very unclear and confusing, where the issues are hidden in the unclearly stated formulation.

I can and only can increase the score to Borderline accept.

**Questions:**

The main questions are left along with the weakness points.


How is the computational efficiency of Algo 1 and how does the number of selected ID samples influence the time?

**Limitations:**

There is a paragraph discussing limitations in the paper, mainly focusing on the general limitation of OOD detection methods. The authors may enhance it by focusing on the proposed techniques, such as how the selected ID sample may influence the performance, and whether selecting more ID samples may influence the computation time.

---

> ### Author Rebuttal · Authors · 2023-08-09
>
>
> Dear Reviewer fiAZ,
>
> We appreciate your time and effort in reviewing our paper. Your feedback provides us with valuable insights into the perception of our work, its strengths, and aspects that require improvement. In the following, we provide the response to your questions:
>
> - How is gradient calculated on the testing samples?
>
> We greatly appreciate your request for clarification. Our methodology involves utilizing an all-one vector as the ground truth, which assumes a uniform distribution for the target data. As a result, we compute the cross-entropy loss by contrasting the model's predicted softmax probability with a uniform vector employed as the target. It's noteworthy that our approach aligns with the GradNorm [1] methodology.
>
> - Subspace computation
>
> Please refer to subsection "Pre-trained Network subspace Computation" in section 3, page 4 of the paper that explains Algorithm 1 in details. As it is presented in line 173, section 3 of the paper:
> "we construct a representation matrix denoted as $R_{ID}^L =[x_{1}^L, x_{2}^L, ..., x_{n}^L ]$, which concatenates $n$ representations obtained from the network's last layer ($L$) through the forward pass of $n$ randomly selected samples (a small subset, $n \ll N$) of the ID data".
>
> - How randomness is done?
>
> We appreciate your highlighting the concern regarding our randomness computations. To ensure a comprehensive and equitable comparison, we incorporate two levels of randomness: trial-level and sample-level.
>
> For our experimental investigations, we adopt an average-results-over-5-runs approach. In each run, distinct random seeds are employed to select random samples from each class, generating small subsets of in-distribution data. Subsequently, we compute the subspace of the pre-trained network based on these subsets. The OOD scores of the test data are then calculated, and FPR95 and AUROC scores are derived. This process is repeated five times, and the average of these five runs is reported as the final score. As documented in our paper, each subspace is computed using ten random samples per class in the ImageNet benchmark and five random samples per class in CIFAR benchmarks.
>
> In response to your feedback, we have taken into account the standard deviations estimated across these five runs. These deviations are presented in Tables 5, 6, and 7 within the attached PDF file, and we incorporate this information in our revised version.
>
>
>
>  - How is the computational efficiency of Algo 1 and how does the number of selected ID samples influence the time?
>
>       -  Our algorithm's runtime for OOD detection remains unaffected by the number of ID samples, as these samples are "only used once" for the subspace computation. In addition, computing the pre-trained network subspace merely involves passing these small subsets of data through the forward phase, without requiring any back-propagation. Consequently, the pre-processing computational complexity remains low, contributing to its efficiency.
>
>       - Our experimental studies present that the number of ID samples leveraged for subspace computation does not significantly influence the algorithm's performance as presented in section D of Appendix. To address your concern, we have conducted additional experiments to assess the impact of the number of ID samples on GradOrth's performance on the ImageNet dataset (in addition to our previously reported experiments on CIFAR dataset in section D of Appendix). You can find the results in Table 3 in the attached PDF file, and for further details, please refer to Appendix, Section D.
>
>       - GradOrth's time complexity is akin to that of GradNorm [1], except for the additional pre-processing step to compute the pre-trained network subspace.
> -  The authors may consider analyzing and evaluating the distribution of the scores of ID and OOD samples, as in previous works.
>
> We greatly appreciate your insightful suggestion. Unfortunately, due to space constraints, we were unable to include the plots at this juncture. However, please rest assured that we are committed to incorporating these plots in our work to visually present the distribution of scores for both in-distribution (ID) and out-of-distribution (OOD) samples.
>
>
> [1] Rui Huang, Andrew Geng, and Yixuan Li. On the importance of gradients for detecting distributional shifts in the wild. Advances in Neural Information Processing Systems, 34:677–689, 2021.

---

> > ### Author Response · Authors · 2023-08-18
> >
> > Dear Reviewer fiAZ,
> >
> > As the discussion period ends soon, we just wanted to check if the response clarified your questions.
> >
> > Thanks again for your constructive feedback.
> >
> > Best,
> >
> > Authors

---

> > ### Comment · Reviewer_fiAZ · 2023-08-21
> >
> > Thanks for the response, which fairly addressed my concerns mentioned in the review. The authors need to fix the problem in the next version of the draft. The representation in the current version is very unclear, as mentioned in the review.
> >
> > The proposed method does make sense and performs slightly better than the main competitors, without supersize. It is a combination of several existing methods, such as GradNorm and the projection-based methods of CL. I didn't mention this issue since the representation in the original paper is very unclear and confusing, where the issues are hidden in the unclearly stated formulation.
> >
> > I can and only can increase the score to Borderline accept.

---

### Official Review · Reviewer_ne6G · 2023-07-09

**Soundness:** 3 good
**Presentation:** 2 fair
**Contribution:** 3 good
**Rating:** 6
**Confidence:** 4

**Summary:**

A method is proposed to perform out-of-distribution detection on a pre-trained classifier. The approach used by the authors is to utilize a low-rank approximation of the parameter space for in-distribution data. Gradients for in-distribution samples are more likely to be projected directly onto the parameter subspace, whereas OOD gradients are more likely to be orthogonal and have small projection values. The proposed method is simple to implement and results in impressive performance compared to other methods for out-of-distribution detection

**Strengths:**

1) The proposed method has low computational complexity and be easily applied to any pre-trained model.

2) Impressive performance on a comprehensive benchmark.

3) Good ablations performed to investigate robustness of the method and effects of different method components.

**Weaknesses:**

1) The specified threshold $\epsilon_{th}$ is not discussed in very much detail. It is unclear if the model performance depends on heavily tuning this value.

2) Experiments are performed on data with significant differences between ID an OOD data. It would be valuable to assess results with CIFAR10 and CIFAR100 as an ID / OOD pairing as in "Exploring the Limits of Out-of-Distribution Detection".

3) Writing and notation could be improved in section 2 and 3 (see questions for more comments)

**Questions:**

1)  How is this threshold $\epsilon_{th}$ selected, it is constant across datasets? How robust is the model as this parameter is varied? It would beneficial to include the value of $k$ in the results table descriptions.

2) What is the effect of batch statistics on the performance of your method? For example, ResNet uses batch normalization, meaning that the ratio of ID and OOD samples during test time will have an effect of the model predictions. How are batch statistics controlled in your experimentation?

Notation:

2) In section 2.3, is there a reason why $O(\mathbf{x})$ was selected? It might be worth changing this to a different letter to prevent confusion with big O notation.

3) In section 2.4 the typical notation is to represent vectors with lowercase boldface. Is there a reason for using alternative notation?

4) In section 3 the acronym ID and OOD are introduced multiple times in this section.

**Limitations:**

Limitations are not discussed in detail. It is recommended to include a section in the Appendix to discuss this.

---

> ### Author Rebuttal · Authors · 2023-08-09
>
>
> Dear Reviewer ne6G,
>
> Thank you for taking the time to evaluate our proposed method. We appreciate your feedback and insights. Your assessment provides us with valuable insights into the strengths and areas for improvement of our approach.
> we want to assure you that we address your questions and concerns in our revision. Please find our responses regarding your questions in the following:
>
> - Assess results with CIFAR10 and CIFAR100 as an ID / OOD pairing:
>
> Thank you for your valuable suggestion in conducting further experimental investigations pertaining to near-ID OOD detection and semantic outliers. Our empirical findings have consistently demonstrated remarkable and reliable performance across both near-ID OOD detection and semantic outlier scenarios.
>
> In our comprehensive near-ID experiments, outlined in Table 1 of the attached PDF, our approach showcased notably robust results, particularly evident within the CIFAR datasets. Furthermore, when applied to the SVHN dataset, our GradOrth approach displayed competitive performance, securing a position among the top three methods. These experiments were executed within the experimental parameters specified in [1].
>
> Furthermore, our GradOrth method has consistently displayed exceptional and unwavering performance in evaluating semantic outliers. As depicted in Table 2, GradOrth outperforms other baseline techniques in handling semantic shifts (S), a notably challenging OOD task, particularly on the real-B dataset. It is worth noting that the performance of GradOrth rivals that of the Mahalanobis method while maintaining a lower computational complexity.
>
>  - Experiments evaluating the role of $\epsilon_{th}$:
>
> We appreciate your request for clarification.  $\epsilon_{th}$ plays an important role but the method works well if its value is high (close to 1). The hyperparameter $\epsilon_{th}$, confined to the range $(0,1)$, serves as a threshold that influences the selection of the value of $k$. The initial $k$ column vectors within matrix $U$ encompass the most pivotal input (representation) space for the pre-trained network. We conducted an experiment to assess the impact of $\epsilon_{th}$ on GradOrth's performance, with results outlined in Table 2 in the attached PDF. Notably, values of $\epsilon_{th}$ near 1 exhibit substantial effectiveness, and we empirically set it to $0.97$.
>
> For a more comprehensive understanding of the rationale behind selecting the value of $\epsilon_{th}$, we kindly direct you to our response to Reviewer (MCXS).
>
> - Effect of Batch normalization on the method
>
> The batch statistics does not affect GradOrth OOD score computations. GradOrth computes per-sample gradient and leverages it to compute the OOD score of the sample.  GradNorm also computes per-sample gradient for OOD detection which avoids the the batch statistics impact on OODness score.
> However, in case of using specific libraries or functions to compute the gradient batch-wise, we can address the issue is using "global statistics" during test time rather than batch statistics. Global statistics refer to the overall mean and variance of the the training dataset, and they provide a more robust representation of the data distribution. By using global statistics during test time, the model is less sensitive to the batch composition.
>
> - Prevent confusion with big O notation.
>
>  We appreciate your valuable observation. While many existing studies utilize the notation $g(x)$ to denote the OOD function, we have recognized a potential conflict with our representation of the gradient for sample $x$, denoted as $g(x)$. In our revision, we will adjust the notation to prevent any ambiguity and to ensure compatibility with complexity scores (big O notation).
>
> [1] Chandramouli Shama Sastry and Sageev Oore. Detecting out-of-distribution examples with gram234
> matrices. In International Conference on Machine Learning, pages 8491–8501. PMLR, 2020.235

---

### Author Rebuttal · Authors · 2023-08-10


Dear Reviewers,

We would like to express our sincere gratitude for your diligent and insightful reviews of our paper. Your thoughtful evaluation and constructive feedback are invaluable to us as we strive to enhance the quality and impact of our work.
 We definitely consider your insightful suggestions and comments in our revision.

In the attached PDF file, we have provided more experimental results raised by reviewers.

In addition to our extensive experiment reported in the paper, we run more experiments considering Near-ID OOD detection experiments.  Our empirical findings have consistently demonstrated remarkable and reliable performance across both near-ID OOD detection and semantic outlier scenarios.

In our comprehensive near-ID experiments, outlined in Table [1] of the attached PDF, our approach showcased notably robust results, particularly evident within the CIFAR datasets. Furthermore, when applied to the SVHN dataset, our GradOrth approach displayed competitive performance, securing a position among the top three methods. These meticulous experiments were executed within the experimental parameters specified in [1].

Furthermore, our GradOrth method has consistently displayed exceptional and unwavering performance in evaluating semantic outliers. As depicted in Table [2], GradOrth outperforms other baseline techniques in handling semantic shifts (S), a notably challenging OOD task, particularly on the real-B dataset. It is worth noting that the performance of GradOrth rivals that of the Mahalanobis method while maintaining a lower computational complexity.

[1] Chandramouli Shama Sastry and Sageev Oore. Detecting out-of-distribution examples with gram234 matrices. In International Conference on Machine Learning, pages 8491–8501. PMLR, 2020.235

---

### Decision · Program_Chairs · 2023-09-21

**Decision:**

Accept (poster)

**Comment:**

This paper proposes GradOrth, a simple post-hoc OOD detection method based on gradient information. Specifically it leverages a selected lower-rank subspace of the gradient most relevant for OOD detection. The proposed method is simple to implement and achieves improved performance compared to other methods for out-of-distribution detection. The idea presented in the paper is interesting and novel. The empirical evaluations are comprehensive.  Many important details are missing and should be included in the revision.